# Text-to-Code Generation for Modular Building Layouts in Building Information Modeling

**Yinyi Wei**
The University of Hong Kong
wyyy@connect.hku.hk

**Xiao Li**[*]
The University of Hong Kong
shell.x.li@hku.hk

## Abstract

We present Text2MBL, a text-to-code generation framework that generates executable Building Information Modeling (BIM) code directly from textual descriptions of modular building layout (MBL) design. Unlike conventional layout generation approaches that operate in 2D space, Text2MBL produces fully parametric, semantically rich BIM layouts through on-the-fly code instantiation. To address MBLs' unique challenges due to their hierarchical three-tier structure: modules (physical building blocks), units (self-contained dwellings), and rooms (functional spaces), we developed an object-oriented code architecture and fine-tuned large language models to output structured action sequences in code format. To train and evaluate the framework, we curated a dataset of paired descriptions and ground truth layouts drawn from real-world modular housing projects. Performance was assessed using metrics for executable validity, semantic fidelity, and geometric consistency. By tightly unifying natural language understanding with BIM code generation, Text2MBL establishes a scalable pipeline from high-level conceptual design to automation-ready modular construction workflows. Our implementation is available at https://github.com/CI3LAB/Text2MBL.

## 1 Introduction

Modular construction has come to the fore in industrial development, replacing traditional construction with standardized 3D volumetric units manufactured off-site and subsequently assembled on-site [14]. This paradigm shift introduces new challenges in building layout design, as traditional spatial configurations must be reinterpreted within the constraints imposed by modular systems. Designing within this confined solution space resembles "dancing with shackles on," where designers must simultaneously address diverse user requirements while operating within a highly restricted design environment [26, 1]. The factory-based production of modular construction draw strong parallels with industrial manufacturing processes, following a trajectory toward mass customization, which seeks to reconcile standardization with the demand for individualized preferences [10, 36, 35]. Consequently, design decision-making is no longer designer-exclusive but user-inclusive.

To facilitate more intuitive and accessible user interaction during the early design phase, text has increasingly been adopted as an input modality in industries such as manufacturing and construction [19, 11, 20]. Recent text-based input methods allow users to express design intentions or preferences, supporting diverse downstream applications, e.g., CAD modeling [19], electronic device [18], and graphic layout design [21]. Although text-based methods have shown promise in generating traditional building layouts [20, 32], extending these approaches to modular building layout (MBL) design remains non-trivial. Unlike conventional practices, modular construction is governed by the "module" concept, adding a layer of complexity.

---

[*]Corresponding Author

39th Conference on Neural Information Processing Systems (NeurIPS 2025).

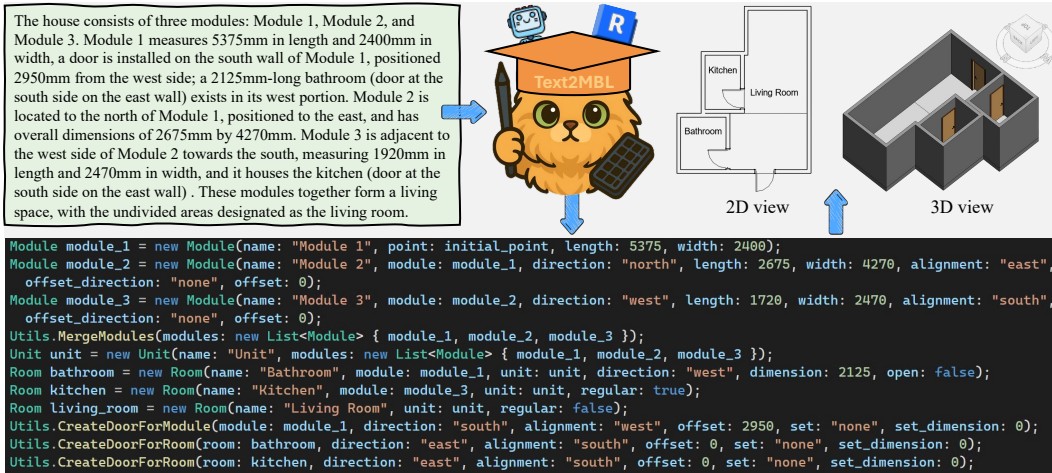

The house consists of three modules: Module 1, Module 2, and Module 3. Module 1 measures 5375mm in length and 2400mm in width, a door is installed on the south wall of Module 1, positioned 2950mm from the west side; a 2125mm-long bathroom (door at the south side on the east wall) exists in its west portion. Module 2 is located to the north of Module 1, positioned to the east, and has overall dimensions of 2675mm by 4270mm. Module 3 is adjacent to the west side of Module 2 towards the south, measuring 1920mm in length and 2470mm in width, and it houses the kitchen (door at the south side on the east wall) . These modules together form a living space, with the undivided areas designated as the living room.

```
Module module_1 = new Module(name: "Module 1", point: initial_point, length: 5375, width: 2400);
Module module_2 = new Module(name: "Module 2", module: module_1, direction: "north", length: 2675, width: 4270, alignment: "east",
  offset_direction: "none", offset: 0);
Module module_3 = new Module(name: "Module 3", module: module_2, direction: "west", length: 1720, width: 2470, alignment: "south",
  offset_direction: "none", offset: 0);
Utils.MergeModules(modules: new List<Module> { module_1, module_2, module_3 });
Unit unit = new Unit(name: "Unit", modules: new List<Module> { module_1, module_2, module_3 });
Room bathroom = new Room(name: "Bathroom", module: module_1, unit: unit, direction: "west", dimension: 2125, open: false);
Room kitchen = new Room(name: "Kitchen", module: module_3, unit: unit, regular: true);
Room living_room = new Room(name: "Living Room", unit: unit, regular: false);
Utils.CreateDoorForModule(module: module_1, direction: "south", alignment: "west", offset: 2950, set: "none", set_dimension: 0);
Utils.CreateDoorForRoom(room: bathroom, direction: "east", alignment: "south", offset: 0, set: "none", set_dimension: 0);
Utils.CreateDoorForRoom(room: kitchen, direction: "east", alignment: "south", offset: 0, set: "none", set_dimension: 0);
```

Figure 1: Example of Text2MBL.

Another challenge exists between conceptual design and construction workflows regarding output format compatibility. While previous research emphasized image-based design generation, the direct production of Building Information Modeling (BIM)-based designs remains underexplored. Unlike image representation, BIM encodes semantically rich and structured information that supports not only visualization but also a wide range of downstream tasks throughout the building lifecycle, e.g., simulation, quantity takeoff, and facility management [30, 33]. However, transitioning the output format from images to BIM introduces additional difficulties, as current approaches typically rely on error-prone post-processing methods that struggle with spatial clashes and semantic incongruity (e.g., raster to vector [23] and IFC-based reconstruction [24]). Although recent work has attempted to generate BIM component coordinates using large language models [11], research has yet to develop methodologies for automated design generation in intricate design contexts, e.g., MBL design.

To address the above challenges, we propose Text2MBL, a framework that translates textual descriptions into BIM-based MBLs through fine-grained, action-level code over sequential time steps. This work focuses on the parametric design setting, wherein users specify precise spatial and geometric requirements through text. By extending BIM's object-oriented nature, Text2MBL hierarchically structures MBL into four classes: Modules, Unit, Room, and Utils. To validate Text2MBL's viability, a proof-of-concept system was implemented in Autodesk Revit [3], a representative BIM platform, where custom classes and functions were developed using the Revit API [4] in C#. This action-based sequential representation delivers three merits: (1) it bridges the modality gap between textual descriptions and BIM-compatible format, as sequential actions (i.e., BIM code) can be automatically executed to produce MBLs in a BIM environment; (2) it eliminates the need for complex geometric reasoning and spatial arrangement computations through abstraction and encapsulation with classes and functions, simplifying the inference process from both user and model perspectives; and (3) it reframes the problem as a sequence-to-sequence generation task, aligning it with established practices in conditional text and code generation. To support training and evaluation, we curated data pairing textual descriptions with action-based instructions in code format from real-world housing projects. Large language models from the Qwen2.5 family [29] were fine-tuned to learn the mapping between text and BIM-based MBLs. We evaluated models from multiple perspectives: executable validity (whether the generated code can be compiled and run in the BIM environment), semantic fidelity (coherence between the input intent and the generated code), and geometric consistency (alignment of the spatial configuration). An illustrative example of the Text2MBL workflow is provided in Fig. 1, where a textual description is translated into a BIM-based MBL.

The contributions of Text2MBL can be summarized as follows:

- We formalize the concepts of BIM-based MBLs following MBL design principles and validate the feasibility through a proof-of-concept implementation in Autodesk Revit.
- We construct a dataset linking textual descriptions with action-based BIM code.

- We formulate the text to BIM-based MBL task as a sequence-to-sequence problem and demonstrate the effectiveness of our approach by fine-tuning large language models.

## 2 Background

**Modular construction** Conventional construction methods have long grappled with persistent problems, e.g., high accident rate, project overrun, and suboptimal quality [25, 28]. Modular construction, characterized by the off-site prefabrication for on-site assembly, has progressively alleviated this predicament [14, 12]. It offers a promising resolution to the classic project management triangle of cost, time, and quality [6]. To provide a theoretical foundation for MBL design, Lin et al. [22] has distilled MBL design principles, including functional space, relationship, and intensity. While comprehensive, current MBL design processes still require extensive expertise and lack mechanisms to incorporate diverse user opinions, hindering mass customization efforts.

**Building information modeling (BIM)** Building Information Modeling (BIM) is a digital, object-oriented representation that captures both the physical and functional aspects of a facility, supporting its entire lifecycle [7]. Compared with traditional CAD, which primarily emphasizes geometric drafting and visualization, BIM extends beyond shapes to incorporate semantic information such as materials, spatial relationships, and lifecycle attributes [8]. Among BIM platforms, Autodesk Revit is one of the most widely used tools in industry and academia. Revit not only provides a modeling environment for creating detailed BIM models but also exposes APIs that allow customization and extension, enabling automated workflows and integration with external applications. While AI-driven design automation has been applied in BIM workflows [40, 34], generating semantically valid BIM models directly from textual descriptions remains largely unexplored.

**Design with textual input** Prior approaches to conditional building layout design have explored pixel-wise generation methods, where each pixel is identified as part of a specific component [27, 31]. While effective for generating artistic pictures, such approaches are ill-suited for precise, constraint-aware design scenarios. Recent advances in layout and design-related domains have emphasized the use of intermediate representations (e.g., component coordinates) to bridge natural language and design outputs precede final rendering in workflows. AutomaTikZ [5] uses TikZ code as an intermediate for generating scientific vector graphics, while DiagrammerGPT [39] employs notions with GPT planning for open-domain diagrams. In 3D design, AnyHome [13] and Open-Universe scene generation [2] utilize scene graphs and program synthesis for indoor spatial design. Similar strategies appear in other domains, including generating electronic devices from text [18], graphic layouts via parse-then-place [21], and sequential CAD models through Text2CAD [19]. Collectively, these works show that code, symbolic grammars, and structured placement plans serve as critical intermediates for controllable and interpretable design synthesis. Motivated by this shift, emerging research in building layout generation from text has proposed generating component-level bounding box coordinates via language models as intermediate representations [9, 20, 32, 11], which offer improved interpretability and adjustability that better accommodate constraint-aware architectural design. However, these methods still require careful management of component ordering to avoid overlaps or inconsistencies. Furthermore, language models often struggle with capturing complex spatial relationships and geometric reasoning.

## 3 Text2MBL

### 3.1 Motivation

The overall framework of Text2MBL is shown in Fig. 2. The MBL generation process is represented as an action sequence that construct a BIM model, encoded using an elaborately designed code architecture. For example, for a two-module MBL, the process begins with the creation of an initial module, followed by the generation of a second module using the first as an anchor. Once the modules are established, individual units are defined within them. Next, specific rooms are assigned based on the existing units and modules, followed by the addition of other architectural elements, e.g., doors. Non-architectural components (e.g., modules) are created through encapsulated functions that define a series of operations to cover essential architectural elements (e.g., walls and floors).

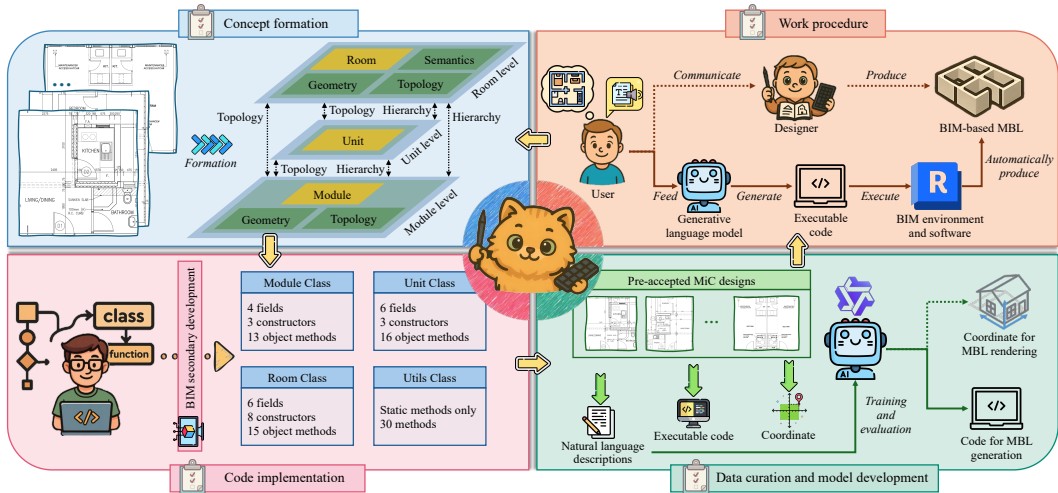

Figure 2: Framework of Text2MBL.

## 3.2 Concept formation

Concepts are first formulated to establish the backbone of the MBL design. We adopt the MBL design principles [22] and its underlying "space without for" theory [15], positing that certain geometric problems are determined not upon the precise shape of individual objects but by their spatial arrangement (see Appendix A for further details). We extend these established principles by systematically analyzing existing MBL examples to support a wider spectrum of MBL scenarios.

Formally, an MBL is defined as $L = \langle \mathcal{M}, \mathcal{U}, \mathcal{R}, \mathcal{E}, \mathcal{A}, \mathcal{C} \rangle$, where $\mathcal{M} = \{m_i\}_{i=1}^{N_\mathcal{M}}$, $\mathcal{U} = \{u_i\}_{i=1}^{N_\mathcal{U}}$, $\mathcal{R} = \{r_i\}_{i=1}^{N_\mathcal{R}}$ denote the sets of modules (i.e., physical building blocks), units (i.e., self-contained dwellings), and rooms (i.e., functional spaces), respectively. Each module and room is associated with specific geometric information (e.g., length and width). Each room $r_i$ is assigned a semantic label indicating its functionality (e.g., kitchen and bathroom). The set $\mathcal{E} = \{e_i\}_{i=1}^{N_\mathcal{E}}$ includes architectural elements such as doors and holes (i.e., passages). The spatial relationships among components are encoded by: the adjacency matrices $\mathcal{A} = \{\mathcal{A}^\mathcal{M}, \mathcal{A}^\mathcal{R}\}$ and the connectivity matrices $\mathcal{C} = \{\mathcal{C}^\mathcal{M}, \mathcal{C}^\mathcal{R}\}$. Adjacency matrices quantify the spatial proximity between modules or rooms based on boundary contact. Connectivity matrices describe the accessibility between spaces, determined by architectural elements (e.g., doors and open walls).

Besides adjacency and connectivity, MBL introduces an additional relationship termed conjoint, capturing the co-location of rooms within the same module. Two rooms $r_i$ and $r_j$ are said to be conjoint if there exists a module $m \in \mathcal{M}$ such that $r_i \subseteq m$ and $r_j \subseteq m$. This relation provides auxiliary topological insight into how spatial components are organized within modular structures.

An MBL must satisfy hierarchical containment constraints. First, each unit must be fully enclosed within a union of one or more modules: $\forall u_j \in \mathcal{U}, u_j \subseteq \bigcup_{m \in \mathcal{M}_j} m$, where $\mathcal{M}_j \subseteq \mathcal{M}$. Second, each room must be nested within a specific unit: $\forall r_k \in \mathcal{R}, r_k \subseteq u_j$ for some $u_j \in \mathcal{U}$. These constraints serve as foundational rules and reflect the hierarchical structure intrinsic to modular architecture.

## 3.3 Code implementation

Based on the established concepts, we developed a proof-of-concept system by extending an existing BIM platform, Autodesk Revit, through its official C# API to ensure seamlessly object-oriented integration. The implementation abstracts complex operational logic into modularized classes and functions, exposing only high-level interfaces suitable for invocation by fine-tuned language models.

The core of the system is three classes: Module, Unit, and Room, with each encapsulating attributes to reflect its role in MBL designs. These classes support multiple constructors, allowing flexible

Table 1: Actions in Text2MBL, exemplified via relevant MBLs, code snippets, and BIM formats.

| Action | Method | MBL (E.g.) | Code (E.g.) | BIM (E.g.) |
|---|---|---|---|---|
| **Module creation** | Absolute coordinate | | *Module module = new Module(name: "Module", point: initial_point, length: 2800, width: 6880);* | |
| | Relative position | | *Module module_3 = new Module(name: "Module 3", module: module_1, direction: "south", length: 2240, width: 1620, alignment: "east", offset_direction: "west", offset: 2000);* | |
| **Module operation** | Module split | | *List<Module> new_modules = Utils.SplitModule(module: module_2, direction: "west-east", ratio: 0.5);\nModule module_2_north = new_modules[0];\nModule module_2_south = new_modules[1];* | |
| | Module merging | | *Utils.MergeModules(modules: new List<Module> module_1, module_2 );* | |
| **Unit initialization** | Combination of modules | | *Unit unit_1 = new Unit(name: "Unit 1", modules: new List<Module> module_1, module_2_north );* | |
| | Combination of directional modules | | *Unit unit_1 = new Unit(name: "Unit 1", modules: new List<Module> module_1, module_2, module_3 , direction: "north", dimensions: new List<double> 5800, 5800, 5800 );* | |
| **Room assignment** | Module/Unit-based | | *Room living_room = new Room(name: "Living Room", module: module, unit: unit, regular: false);* | |
| | Diretion-oriented | | *Room kitchen = new Room(name: "Kitchen", module: module, unit: unit, direction: "south", dimension: 1800, open: true);* | |
| | Corner-oriented | | *Room kitchen = new Room(name: "Kitchen", module: module, unit: unit, corner: "southwest", length: 1600, width: 1200, offset_direction: "none", offset: 0, open: true);* | |
| | Relative | | *Room kitchen = new Room(name: "Kitchen", unit: unit, room: bathroom, direction: "east", length: 1640, width: 1220, alignment: "north", offset_direction: "none", offset: 0, open: false);* | |
| **Element placement** | Door | | *Utils.CreateDoorForRoom(room: bedroom_3, direction: "west", alignment: "south", offset: 0, set: "in", set_dimension: 600);* | |
| | Hole | | *Utils.CreateHole(module: module_2, direction: "north", alignment: "none", offset: 0, dimension: 2000);* | |

instantiation based on diverse textual input. Each class provides object methods for fundamental geometric and structural operations, e.g., retrieving geometric features.

Beyond class-specific methods, a utility (Utils) class was implemented to provide static methods for both geometrical operations (e.g., midpoint calculation and concave polygon detection) and functional processes for MBL components (e.g., creating doors and holes, splitting or merging modules).

Table 1 presents representative atomic actions defined in Text2MBL, along with illustrative examples and corresponding code interfaces. Additional implementation details are provided in Appendix B.

## 3.4 Data curation and model development

After determining the input and output formats (i.e., textual descriptions and corresponding code-based action sequences), we constructed our dataset by collecting pre-accepted modular construction designs from official sources[2]. Each design instance was manually annotated with two constituents: (1) executable BIM code under the developed architecture that generates valid MBL models automatically and (2) corresponding textual descriptions that articulate the user's design intent following parametric design.

---

[2]https://www.bd.gov.hk/en/resources/codes-and-references/modular-integrated-construction/mic_acceptedList.html

From this process, we curated a dataset comprising 198 unique MBL designs. Each MBL instance was first annotated with its associated code-based action sequence, compiled and verified for correctness. Then, two distinct textual descriptions were independently written for each MBL to simulate diverse user expressions. The two different descriptions for a MBL design follow different narrative sequences and structures, such as describing the components in various order. Two coding styles are considered: one utilizing named arguments (i.e., explicitly specifying the semantic roles of each parameter) and the other using positional arguments (i.e., relying solely on the order of parameters). The resulting dataset is formally defined as $\mathcal{D} = \left\{ \left( d_i^a, d_i^b, c_i^{\text{name}}, c_i^{\text{pos}} \right) \mid i = 1, 2, \ldots, N_{\mathcal{D}} \right\}$, where $d_i^a$ and $d_i^b$ are the two distinct textual descriptions, $c_i^{\text{name}}$ and $c_i^{\text{pos}}$ are the corresponding code following named argument and positional argument formats, respectively.

Due to MBL's recent inception, there is a paucity of publicly available training data specific to MBL design tasks. To alleviate data deficiency and improve generalizability, we adopt synthetic data generation in both partial and full manner.

In the partially synthetic setting, existing golden code samples are leveraged for generating textual descriptions. A mapping function $\mathcal{T}$ is defined to convert each code sample $c$ to a corresponding textual description $d^p$, such that $d^p = \mathcal{T}(c)$. We explored two realizations of $\mathcal{T}$: a template-based approach and a model-based approach. In the template-based variant, a bank of five linguistically diverse templates was defined for each action scenario (e.g., creating a module with varying parameter combinations). During generation, one template was sampled uniformly at random for each action. The model-based variant employed a proprietary model (i.e., gpt-4.1-mini) to convert code into textual descriptions with the prompt used in Appendix J.

In the fully synthetic setting, both code samples and associated descriptions are generated de novo. A code sample $c^f$ is synthesized based on predefined grammar rules and parameter configurations that reflect plausible usage scenarios observed in realistic data. Each generated instance is then converted into a textual description $d^f$ using the mapping function described above, yielding $d^f = \mathcal{T}(c^f)$.

We seek a parameterized language model $p_\theta(c \mid d)$ that, given a description $d$, produces the code-based action sequence $c$ of a BIM-based MBL design $L$. Concretely: $\hat{c} = \arg\max_c p_\theta(c \mid d)$.

## 4 Experiment

Given the lack of existing benchmarks for this problem, our experimental objectives is twofold: (1) to quantify the performance of our proposed code-driven generation approach compared to conventional coordinate-driven generation approach at hand, and (2) to identify pathways for continual improvement, e.g., model selection and synthetic data generation.

### 4.1 Experimental setup

For the 198 MBL designs (396 pairs of descriptions and code), we partitioned them into training, development, and test sets using a 7:1:2 split, resulting in 138, 20, and 40 designs (276, 40, and 80 pairs). In the partially synthetic setting, we generated 10 descriptions per design using both template- and model-based approaches, producing 1,380 new descriptions per method. For the fully synthetic setting, we synthesized 3,000 unique code sequences from scratch and converted them into textual descriptions using the same two approaches. To compare with coordinate-driven generation methods [3], we derived bounding box coordinates of MBL designs. The output coordinate sequence was structured hierarchically, consisting of three segments (i.e., modules, units, and rooms): `MODULE:\n[module seq]\nUnit:\n[unit seq]\nRoom:\n[room seq]`. Within each segment, every component is described by its bottom-left corner, length, and width. For example, a module can be represented as `[Module 1|x=0|y=0|length=3100.0|width=5420.0]`. Detailed statistics (e.g., the number of tokens of descriptions and code) are provided in Appendix C.

We fine-tuned models from the Qwen2.5 family [29], leveraging their strong capabilities as backbone models. To investigate the effects of model scale and specialization, we conducted experiments using variants of Qwen2.5 across different sizes (0.5B, 1.5B, 3B, and 7B) and domains, including the vanilla series [37], the coder series [17], and the math series [38]. To ensure a reliable comparison,

---

[3]We omit naming specific coordinate-driven methods (e.g., Text2BIM or Tell2Design) in experimental comparisons to highlight the fundamental paradigm difference between coordinate- and code-driven approaches

Table 2: Geometric consistency using different models and output formats. Positional arguments are used. Mean values over five views are reported in this and subsequent tables and figures.

| Model | IoU (%) | Module IoU (%) | Unit IoU (%) | Room IoU (%) |
|---|---|---|---|---|
| Qwen2.5-Instruct | | | | |
| 1.5B - Coordinate | 78.13 | 86.98 | 80.27 | 69.85 |
| 1.5B - Code | 91.65 | 95.47 | 93.79 | 86.79 |
| 7B - Coordinate | 85.38 | 90.35 | 90.39 | 77.33 |
| 7B - Code | 95.83 | 98.51 | 98.11 | 91.64 |
| Qwen2.5-Coder-Instruct | | | | |
| 1.5B - Coordinate | 78.33 | 87.71 | 80.20 | 69.85 |
| 1.5B - Code | 94.19 | 97.01 | 95.59 | 90.45 |
| 7B - Coordinate | 84.64 | 90.12 | 88.18 | 77.53 |
| 7B - Code | 98.43 | 98.78 | 99.19 | 97.37 |
| Qwen2.5-Math-Instruct | | | | |
| 1.5B - Coordinate | 78.61 | 87.03 | 85.09 | 67.57 |
| 1.5B - Code | 94.18 | 97.49 | 97.17 | 89.06 |
| 7B - Coordinate | 85.34 | 90.32 | 89.51 | 78.02 |
| 7B - Code | 94.86 | 98.07 | 97.46 | 89.86 |

Figure 3: Component-level performance on compile rate, pass rate, instance F1, argument F1, and IoU for modules, units, rooms, and elements, using Qwen2.5-Coder-3B with positional arguments.

each experiment was conducted five times using different random seeds. Results are presented as mean values across all trials. More experimental details are presented in Appendix D.

We evaluated models along three dimensions: executable validity, semantic fidelity, and geometric consistency. To assess executable validity, we measured whether the generated code compiles successfully (compile rate) and produces functionally correct outputs (pass rate). Given that crucial design information is often embedded in code expression such as class names, function names, and their arguments, we evaluate semantic fidelity using standard information extraction metrics: F1 score. We consider two granularities: (1) instance-level with instance F1, which measures whether a complete code line representing a design action is correctly generated, and (2) argument-level using argument F1, which focuses on the correctness of individual arguments within such instances. To quantify geometric consistency between the generated and reference MBL designs, we employ the Intersection over Union (IoU) metric. IoU measures the degree of spatial overlap between two bounding boxes, i.e., the ratio of the area of their intersection to the area of their union.

## 4.2 Experimental results

**Output formats: code or coordinate** We first evaluate model performance across different output formats, i.e., code and coordinate. As shown in Table 2, the code-driven approach demonstrated superior geometric consistency in generating MBLs, stemming from the relative positioning employed in the code architecture instead of absolute coordinates. Moreover, we note that geometric consistency metrics only quantify the degree of bounding box overlap and may not fully capture critical factors encountered in real-world MBL scenarios, e.g., element placement, clash avoidance, component ordering, and support for irregular geometries. Text2MBL addresses these challenges by abstracting complex design operations into high-level interfaces, facilitating more robust and interpretable MBL

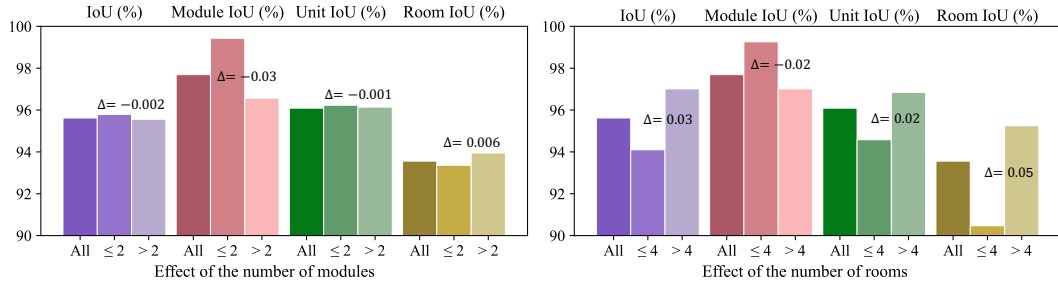

Figure 4: Effect of the number of modules and rooms on geometric consistency metrics, using Qwen2.5-Coder-3B with positional arguments-based code output. Relative transformations ($\Delta$) are computed from groups with fewer components to those with higher components counts.

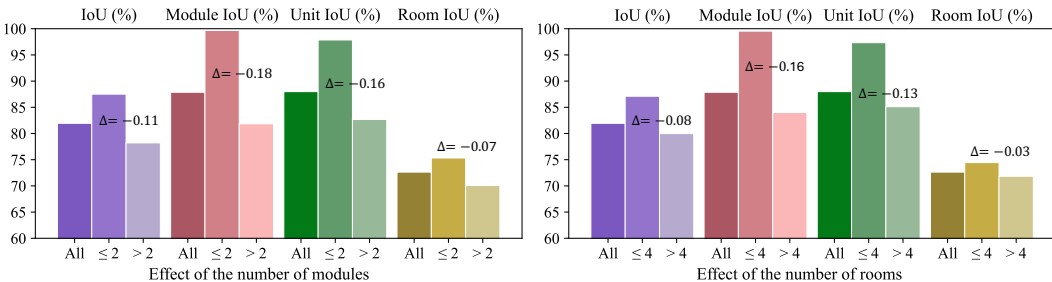

Figure 5: Effect of the number of modules and rooms on geometric consistency metrics, using Qwen2.5-Coder-3B with coordinate sequence output. Relative transformations ($\Delta$) are computed from groups with fewer components to those with higher components counts.

generation in the construction workflows. Additionally, Appendix F presents the performance of representative proprietary models, highlighting the potential of text-to-code generation for MBLs.

**Component-level performance** To gain deeper insights into the performance of this text-to-code generation task, we further report fine-grained, component-level results in terms of compile rate, pass rate, instance F1, argument F1, and IoU for modules, units, rooms, and elements. Fig. 3 revealed that: (1) The semantic fidelity metrics (i.e., instance F1 and argument F1 scores) achieved better results than the pass rate metric, suggesting that models were more effective at extracting argument information than at producing perfectly executable code without deviations. The observation offers inspirations in practical deployment scenarios that extracted arguments can be leveraged to guide post-generation correction, enhancing the overall robustness and usability in real-world applications. (2) The pass rates for modules and rooms are notably lower than those for units and elements, likely due to their increased operational complexity and the higher number of constraints involved. For other metrics, the performance gap among components is less pronounced. As a complement, Appendix E provides more results across various metrics covering different model series and argument types.

**Component number effect** We then group designs based on the number of modules and rooms, aiming to reveal the relative performance trends as the number of components increases. The results from code output and coordinate sequence output are illustrated in Fig. 4 and Fig. 5, respectively. We observe that increasing the number of components led to a sharper performance decline in the coordinate sequence output compared to the code output. This difference can be attribute to the hierarchically designed code architecture, which replaces spatial reasoning with semantic understanding relying on relative position, delivering more robust results. Additional details and discussions can refer to Appendix G.

**Low resource-scenarios and data integration** Though the experiments demonstrate the effectiveness and potential of the Text2MBL framework, a critical issue to the broader deployment is

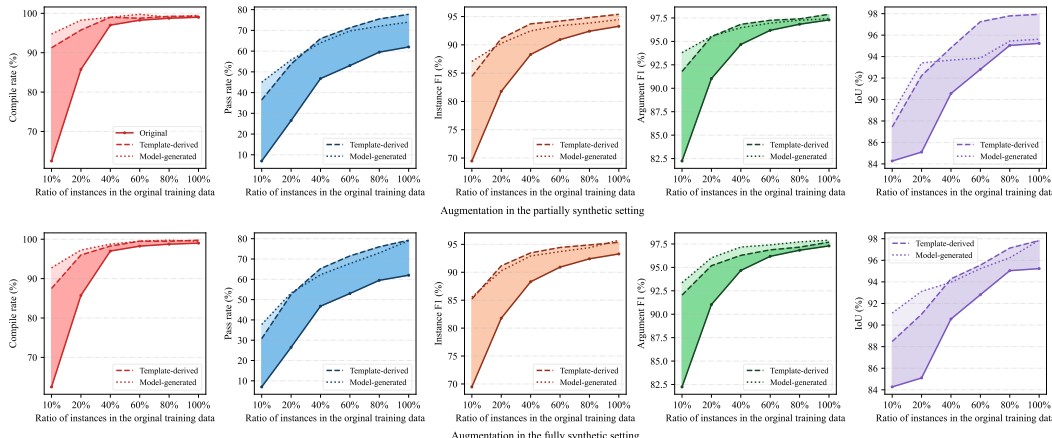

Figure 6: Ratios of original training data augmented with synthetic data, using Qwen2.5-Coder-3B with positional arguments. The corresponding performance improvements for metrics are highlighted.

the lack of sufficient training data, mainly due to two factors: the scarcity of accessible MBL data and the high cost of annotation. To alleviate this issue and facilitate wider application of our framework, we adopted a synthetic data generation strategy as described in Section 3.4. Though the synthetic data face issues such as rigid description logic and has demonstrated inferior performance as revealed in Appendix I. We repurpose the synthetic data for augmentation rather than standalone training. Specifically, we mixed the original training data with synthetic data for model training[4]. As shown in Fig. 6, this mixed strategy yielded substantial performance gains, especially under extreme low-resource conditions (e.g., only 10% original data available). Though the fully synthetic data may introduced noise and inconsistencies in the fabricated code, the integration of such data is still beneficial. While model-based descriptions are more natural than those from the template-based approach, their relative performance gains were comparable and, in some cases, even inferior. This demonstrates that, while using synthetic data alone is insufficient, leveraging it in an augmented manner can significantly enhance performance.

**Abstract and conceptual instructions** To assess the generalization capabilities of fine-tuned models, we conducted experiments using abstract and conceptual instructions. Specifically, we transformed detailed instructions into a skeletonized format: "Generate a layout with $n_1$ module, $n_2$ unit, $n_3$ living room, $n_4$ bathroom, $n_5$ bedroom, $n_6$ kitchen." Model performance was evaluated using two metrics: the compile rate, which reflects the successful generation of executable code, and the extraction F1 score, which quantifies the accuracy of extracting and mentioning modules, units, and rooms.

The empirical results reported in Table 3 demonstrated basic generalization from detailed to abstract instructions, yet sill warranting further in-

Table 3: Results with abstract instructions using various models and argument types.

| Argument | Compile (%) | Extraction F1 (%) |
|---|---|---|
| Qwen2.5-Coder-3B | | |
| Named | 68.75 | 99.05 |
| Positional | 45.00 | 98.36 |
| Qwen2.5-Coder-7B | | |
| Named | 92.5 | 100.0 |
| Positional | 95.00 | 99.53 |
| ChatGPT (gpt-4.1-mini) | | |
| Named | 81.25 | 92.91 |
| Positional | 78.75 | 91.96 |
| ChatGPT (gpt-4.1) | | |
| Named | 87.50 | 96.11 |
| Positional | 82.50 | 94.58 |

vestigation for smaller models and general-purpose proprietary models. Models provided with named arguments generally achieved higher performance, benefiting from the greater specificity and structure afforded by named arguments for general inputs. This suggests that while our approach can handle a

---

[4]We have tried pretraining on synthetic data followed by fine-tuning on original data, but observed no improvement and sometimes worse performance.

degree of abstraction, further improvements are necessary to robustly support the full granularities of user input styles.

## 4.3 Error analysis

We carefully analyzed the generated results and summarized several common types of errors made: (1) wrong argument order, (2) incorrect code structure or sequencing; (3) hallucinated or invalid arguments; (4) hallucination of of non-existent functions; and (5) invocation of incorrect functions.

While these errors highlight the current limitations in fully accurate code generation from textual descriptions, practical systems should incorporate post-generation validation and correction mechanisms to ensure executable and user-aligned outputs, as discussed in Appendix H.

## 5 Discussion

While our work contributes significantly to the automation of BIM design tasks and inspire related work in both manufacturing and construction industries, several limitations remain.

First, in the text-to-code generation task, we only considered the technical aspects of parametric design, relying on detailed user instructions to guide code synthesis. The current framework does not account for the alignment between generated code and user inputs at varying levels of granularity, nor does it address broader user-centric criteria such as satisfaction or usability. A thorough evaluation of these dimensions would require carefully designed user studies, which we leave for future work.

Second, we employed identical families of architectural elements (e.g., wall, floor, and door) to maintain simplicity and consistency in our developed code architecture. However, expressive designs often demand greater variation in elements properties (e.g., width, material) and their combinations. Addressing this need could involve leveraging richer component libraries and more effective retrieval mechanisms, which we plan to explore in future work.

## 6 Conclusion

In this work, we investigate user-inclusive parametric design in construction workflows, focusing on BIM-based MBL generation from textual descriptions. We identify the challenges posed by MBL intrinsic characteristics (i.e., hierarchy, semantics, geometry, and topology) and the output format requirements (i.e., BIM representations). We frame the problem as a code generation task, where structured code serves as executable action sequences for constructing MBLs in BIM environments. We curated a dataset from real-world modular housing projects and fine-tuned large language models to translate textual descriptions into BIM-compatible code. Empirical evaluation of the Text2MBL framework reveals three key insights: (1) The code-driven paradigm substantially outperforms conventional coordinate-based approaches, underscoring the effectiveness of Text2MBL. (2) Across various evaluation metrics, code-driven models exhibit strong performance, especially in extracting information from inputs, confirming the feasibility of Text2MBL. (3) Although the pass rate remains a challenge, performance can be significantly improved through synthetic data augmentation, demonstrating the potential of Text2MBL. A functional plug-in has been implemented within Autodesk Revit. Future research will focus on developing more robust models capable of processing user inputs at varying levels of granularity, thereby generating more accurate and executable layout code. Moreover, the expressiveness of the generated outputs will be extended to incorporate a broader range of design variations.

## Acknowledgments

The work described in this paper is supported by the Research Grants Council of the Hong Kong SAR of China (Germany/Hong Kong Joint Research Scheme, No. G-HKU502/22), the Guangdong-Hong Kong Technology Cooperation Funding Scheme (TCFS) (Ref no. GHP/321/22SZ), and the Innovation and Technology Fund (ITF) (Ref No. TP/041/24LP).

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

Figure 7: Example demonstrating MBL design principles.

# A    The design principles of modular building layouts

To ensure validity and quality of MBLs, three fundamental factors must be carefully considered: (1) functional space, which refers to the assignment of appropriate spatial functionalities to different areas; (2) relationship, encompassing spatial adjacency, connectivity, and conjoint (co-location) between functional components; and (3) intensity, describing the degree of spatial proximity or strength of association between adjacent spaces. These principles collectively guide the structural and semantic coherence of MBLs.

An illustrative example in Fig. 7 demonstrates how these principles can be modeled through graph-based representations.

# B    Additional details on code architecture

We provide supplementary information regarding the code architecture in this section. The system is developed within a three-dimensional orthogonal coordinate framework (X, Y, Z). Due to the consistent specification of elevation levels and wall height parameters across the design, the computational complexity is significantly reduced onto the two-dimensional (X-Y) plane. Furthermore, architectural elements (i.e., walls, doors, and floors) are instantiated from uniform family templates, serving as a common basis for initial modeling and can be flexibly modified during subsequent stages of the conceptual design phase. For comprehensive implementation details and reproducibility, we refer readers to our code repository.

## B.1    Module class

The Module class represents a primitive physical cell in an MBL design within a Revit document. Each instance encapsulates essential semantic and geometric information, including the module's name, boundary coordinates, level, and associated architectural elements such as walls and floor components. Designed for flexibility, the class offers multiple constructors to support different initialization scenarios, accommodating inputs ranging from explicit boundary points to dimensional parameters or relative positioning based on existing modules.

Specifically, the class supports three construction strategies: (1) initialization via four explicit boundary points defining the module's rectangular geometry; (2) initialization from a single anchor point (i.e., the bottom-left corner) combined with specified length and width values; and (3) relative positioning based on a reference module, where the new module is placed according to a given direction (e.g., north, south) and edge alignment (e.g., east, west).

The class also provides methods to retrieve the module's name, level, boundary points, length, and width. Additionally, it includes methods to get specific boundary points (e.g., southeast, northwest) and walls (e.g., east wall).

To support downstream spatial reasoning and manipulation, the class provides a suite of accessor methods that expose both semantic and geometric properties of the module. Additionally, utility methods are available to extract specific corner coordinates (e.g., southeast, northwest) and associated structural elements, such as directional walls (e.g., the east wall).

## B.2 Unit class

The Unit class is designed as an abstraction layer for organizing collections of modules and rooms within a Revit document, representing a self-contained dwelling space in MBL design. It encapsulates key semantic and geometric information, including a unit's label, spatial boundaries, associated building level, constituent modules, and contained rooms. It provides constructors to initialize a unit based on different input parameters, such as a list of modules, boundary points, or directional placement relative to existing modules.

Specifically, the class supports three construction strategies: (1) instantiation from a list of Module objects, in which the unit's geometry is inferred directly from its constituents; (2) initialization with both a list of Module objects and explicitly defined boundary points, offering more granular control when boundary specifications diverge from module arrangements; and (3) instantiation with directional placement of specific modules, where modules are programmatically arranged in specified directions (i.e., north, south, east, west) with user-defined dimensions. In the third strategy, new boundary points are computed for each module based on directional parameters, adjusted for wall offsets, and augmented with walls when absent.

In addition to accessors for geometric properties and boundary data, the class includes methods to manage room composition within a unit, ensuring semantic, spatial, and hierarchical coherence.

## B.3 Room class

The Room class represents a functional space within a Revit document, defined by its semantic role (e.g., kitchen, bedroom), geometric boundaries, centroid, elevation level, and its parent unit within the modular hierarchy. It provides a range of constructor overloads to create rooms based on different input parameters, such as modules, units, specific dimensions, or relative positions.

To accommodate different design scenarios and user requirements, the Room class offers multiple constructors. In accordance with the hierarchical constraints, the instantiation of a Room object requires the explicit specification of a Room parameter, even if it is being initialized within a module.

First, a room may be instantiated within a specified module. In this case, if the room is considered regular, its geometric boundaries are slightly adjusted to accommodate wall thicknesses. In cases where no enclosing walls are detected, these walls are automatically generated to define the space. The centroid is computed as the midpoint of the defined rectangular boundary, where a Revit room and its corresponding tag are created. If the room is irregular (e.g., occupying residual space within a module), the constructor computes a minimum bounding rectangle for the available area and ensures that the computed center does not overlap with any pre-existing rooms. Similarly, rooms can be instantiated at the unit level, operating analogously to the module-based constructor but derives the geometric boundaries from the unit-level layout.

Secondly, a constructor allows direct specification of spatial parameters, i.e., central location, length, and width. In this case, boundary points are computed based on the provided dimensions and adjusted to accommodate construction details (e.g., wall offsets, wall detection, and wall creation).

In more dynamic scenarios, the class provides directional and corner-based constructors. The directional constructor places a room along a specified cardinal direction (i.e., north, south, east, west) relative to a given module or unit. Boundary points are calculated according to the direction and spatial extent, and walls are generated accordingly. If an open-layout flag is enabled, certain walls are omitted to enable spatial continuity. Similarly, the corner-based constructor enables placement in a specific quadrant (i.e., southeast, southwest, northwest, or northeast), supporting directional offsets and optional open layouts via selective wall removal.

Additionally, a room can be instantiated adjacent to an existing room. This adjacency-aware constructor aligns the new room in a specified direction relative to the reference room, computing boundaries based on alignment, offset, and dimension constraints. As with other constructors, it supports both enclosed and open configurations through wall manipulation.

Beyond initialization, the Room class also provides utility methods for querying, computing, and modifying its spatial and semantic attributes.

## B.4   Utils class

The Utils class provides a comprehensive set of utility functions tailored for managing and manipulating Revit elements (e.g., walls, floors, and doors) and supports a variety of advanced operations for modules (e.g., module splitting, merging, and structural modifications). Serving as a centralized utility layer, this class streamlines geometric calculations, element creation, and spatial transformations in a Revit document.

**Conversion utilities**   To address unit inconsistency between Revit (which uses feet as the standard unit) and architectural design data (typically in millimeters), the class provides conversion functions *Foot2mm* and *Mm2foot*, ensuring consistent unit handling across all geometric and placement operations.

**Geometry calculations**   A range of geometric utilities is included for spatial reasoning. Functions such as *MidPointForLine*, *MidPointForRectangle*, and *MidPointForWall* calculate midpoints for various shapes, supporting tasks like element alignment and central placement. *PointInRectangle* determines whether a given point lies within a rectangle defined by four corners, which is essential for containment checks and spatial validation.

**Wall management**   The *CreateWall* function constructs a wall between two 2D points (in UV coordinates), ensuring it is room-bounding and conforms to a designated wall type. Deletion methods include *DeleteWallGivenPoints* and *DeleteWallGivenWall*, supporting both point-based and object-based wall removal. Retrieval methods such as *GetWallByTwoPoints* and *GetAllWallsByTwoPoints* offer querying of existing walls based on boundary conditions. Additionally, *DeleteAllShortWall-GivenPoints* removes short or malformed walls, preserving geometric consistency for downstream tasks.

**Contour and boundary management**   Higher-order geometric utilities are designed for contour and boundary manipulation, serving as crucial parts in MBL generation and rendering. Functions like *Clockwise* and *FindNext* reorder points into a clockwise sequence, ensuring consistent boundary definitions for merged or modified modules. Simplification routines such as *RemoveCollinearPoints* and *RemoveNonParallelPoints* eliminate redundant or misaligned points to maintain structural clarity. *GetWallPoints* further adjusts boundary definitions to account for wall thickness and the nature of angles (concave or convex), optimizing placement accuracy. To facilitate collision detection and placement validation, *PointInWall* and *PointInOneWall* determine whether a point lies within a wall or specific wall section. *IsConcave* identifies concave angles, which is crucial for handling irregular geometries during module generation and modification.

**Floor management**   Floor creation is supported via the *CreateFloor* method, which constructs a floor based on four boundary points. It uses a default floor type under an identical family to ensure geometric closure for valid floor definitions.

**Door and hole management**   Door and hole creation is essential for spatial connectivity in MBL designs, which are considered evaluative elements in our text to BIM-based MBL generation task. *CreateDoor* inserts a door into a specified wall location, handling alignment and structural adjustments. *CreateDoorForModule* and *CreateDoorForRoom* extend this functionality by allowing directional placement and offset configuration within larger spatial contexts. These methods also support the specification of advanced door profiles, such as recessed or protruding forms. Simplified variants like *CreateDoorOnMidpointForModule* and *CreateDoorOnMidpointForRoom* enable automatic door placement at wall midpoints. Openings (or holes) are created using *CreateHole*, which facilitates inter-module connections by defining voids (i.e., passageways) based on alignment and size parameters.

Table 4: Statistic on the token counts (including average, maximum, and minimum) for different output types, i.e., code with named arguments, positional arguments, and coordinate.

| Output type | No. of token | | |
| --- | --- | --- | --- |
| | Avg. | Max. | Min. |
| Code w/ named argument | 506.9 | 1,444 | 111 |
| Code w/ positional argument | 362.6 | 1,060 | 77 |
| Coordinate | 315.3 | 841 | 95 |

Table 5: Statistic across different splits on the average number of tokens and sentences in descriptions and code snippets (with named arguments), as well as the average counts of MBL components, including modules, units, rooms, and elements (doors and holes).

| Split | Description | | Code | | MBL component | | | |
| --- | --- | --- | --- | --- | --- | --- | --- | --- |
| | Sent. | Token | Sent. | Token | Module | Unit | Room | Element |
| Train | 5.7 | 225.0 | 13.2 | 506.9 | 2.8 | 1.1 | 4.3 | 4.5 |
| Dev | 5.6 | 221.0 | 12.7 | 492.4 | 2.5 | 1.1 | 4.5 | 4.3 |
| Test | 6.2 | 231.5 | 13.5 | 523.1 | 2.7 | 1.2 | 4.6 | 4.7 |
| All | 5.8 | 225.9 | 13.2 | 508.7 | 2.7 | 1.1 | 4.4 | 4.5 |

**Advanced module operations**  In MBL design, there are advanced operations to manage modules, providing flexible and dynamic spatial configuration. *MergeModules* combines multiple modules into a single module by removing internal walls and creating new boundary walls. Conversely, *SplitModule* divides a module into two smaller modules based on a specified direction (i.e., north-south or west-east) and a ratio, offering a dynamic mechanism for adaptive space planning.

## B.5   Framework extension

In this study, we focus on modular building layouts composed of rectangular modules, a design choice motivated by prevailing practices in the architecture, engineering, and construction (AEC) industry. The adoption of rectangular modules aligns with the principles of industrialized construction, particularly design for manufacturing and assembly (DfMA) and efficient transportation logistics, which are critical drivers in contemporary modular construction. Consequently, both our methodology and accompanying codebase are optimized for buildings assembled from rectangular modules, reflecting the predominant approach in real-world projects.

Despite this specific focus, our system is designed to be readily extensible to accommodate a broader range of modular geometries. The code architecture is developed with the principle of low coupling and high cohesion, whereby each class and function encapsulates a well-defined responsibility. This modular architecture enables straightforward extension: for example, although the current Module and Utils classes are tailored to rectangular modules, the architecture supports the seamless integration of additional geometric representations. Non-rectilinear modules, such as those with polygonal or curved shapes, can be incorporated by subclassing the existing Module class (e.g., creating PolygonalModule or CurvedModule subclasses) and implementing the necessary geometric methods (such as PointInPolygon). These extensions can introduce new descriptors and functionalities without disrupting the integrity of the existing rectangular module implementation, thus ensuring both robustness and future adaptability.

## C   Detailed data statistics

This section provides more detailed statistics of the data used.

In current progress, we only consider four primary room types commonly found in MBL designs: living room, bedroom, bathroom, and kitchen. With the flexible code architecture, additional room types can be easily incorporated as needed for future applications.

Table 4 displays the token statistics for different output types, including code and coordinate formats. The coordinate outputs, which encode bounding box information, result in comparatively shorter

Table 6: Statistics on average sentence and token counts for code, along with the average number of MBL components, in the original training set and the fully synthetic data.

| Source | Code | | MBL component | | | |
|--------|------|------|--------|------|------|---------|
| | Sent. | Token | Module | Unit | Room | Element |
| Original | 13.2 | 506.9 | 2.8 | 1.1 | 4.3 | 4.5 |
| Full | 13.8 | 574.3 | 3.0 | 1.7 | 5.2 | 3.3 |

Table 7: Statistic on token counts (including average, maximum, and minimum) for descriptions in original training set and synthetic data, using template- and model-based approaches in both partially and fully synthetic settings.

| Source | Avg. | Max. | Min. |
|--------|------|------|------|
| Original | 225.0 | 605 | 40 |
| Partial (template-based) | 254.5 | 712 | 45 |
| Partial (model-based) | 203.6 | 482 | 64 |
| Full (template-based) | 322.0 | 1,038 | 24 |
| Full (model-based) | 352.1 | 1,031 | 46 |

sequences. In contrast, the code outputs encapsulate more detailed semantic and structural information through arguments, providing more necessary details for high-fidelity BIM model generation.

The statistics for the different dataset splits are presented in Table 5.

Table 6 provides the comparison between ground truth code and generated code in the fully synthetic setting.

The statistical comparison of token counts between the original training data and the synthetic data is illustrated in Table 7. The analysis encompasses both template- and model-based generation approaches under partially and fully synthetic settings. In the partially synthetic setting, the generated textual descriptions closely align with the original data in terms of length, reflecting the constraints imposed by the real input. In contrast, fully synthetic data exhibits significantly longer token sequences, which can be attributed to the increased variability and verbosity introduced by randomly generated code.

## D    Experimental details

To ensure computational feasibility within our hardware constraints, we employed Low-Rank Adaptation (LoRA) [16] for fine-tuning. All linear transformations in the model, including the query-, key-, value-, output-, gate-, up-, and down-projection matrices, were equipped with LoRA adapters. The model operated with a maximum input and output length of 800 and 1,600 tokens, and computations were performed using 16-bit BrainFloat precision. Fine-tuning was conducted using a learning rate of 3e-4 and a batch size of 2; to effectively simulate a batch size of 4, gradient accumulation over 2 steps was applied. Training proceeded for 5 epochs, and model selection was based on the highest evaluation scores observed on the development set, where argument F1 was used for code-driven models and IoU was used for coordinate-driven models. LoRA-specific hyperparameters were set with a rank of 64, a scaling factor (alpha) of 64, and an adapter dropout rate of 0.1. During generation after fine-tuning, the model was restricted to producing at most 1,600 new tokens, utilizing greedy decoding. All experiments were performed on a server equipped with four NVIDIA GeForce RTX 4090 GPUs and a 64-core Intel Xeon Platinum 8370C CPU operating at 2.80 GHz.

## E    Model selection and argument types

This section evaluates model performance across different model series and argument types. From the results summarized in Table 8, two key observations emerge.

Firstly, across different model series, though pretrained on extensive code corpora, the coder series performed comparable to their vanilla counterparts in this code generation task. In contrast, the math

Table 8: Results across executable validity, semantic fidelity, and geometric consistency using different models and various argument types.

| Model | Compile (%) | Pass (%) | Instance F1 (%) | Argument F1 (%) | IoU (%) |
|---|---|---|---|---|---|
| Qwen2.5-Instruct | | | | | |
| 0.5B - Named | 98.25 | 45.50 | 87.96 | 93.33 | 90.51 |
| 0.5B - Positional | 99.00 | 49.25 | 89.28 | 93.66 | 93.02 |
| 1.5B - Named | 92.00 | 63.25 | 92.56 | 95.86 | 95.71 |
| 1.5B - Positional | 98.75 | 62.25 | 91.91 | 96.08 | 91.65 |
| 3B - Named | 96.50 | 73.25 | 94.05 | 96.03 | 95.79 |
| 3B - Positional | 98.75 | 71.75 | 93.98 | 96.35 | 95.87 |
| 7B - Named | 95.75 | 78.25 | 95.53 | 97.41 | 96.07 |
| 7B - Positional | 99.5 0 | 76.00 | 94.96 | 97.48 | 95.83 |
| Qwen2.5-Coder-Instruct | | | | | |
| 0.5B - Named | 95.00 | 49.00 | 88.12 | 93.80 | 89.07 |
| 0.5B - Positional | 99.50 | 47.50 | 87.59 | 93.82 | 88.22 |
| 1.5B - Named | 96.25 | 65.00 | 92.34 | 95.69 | 95.45 |
| 1.5B - Positional | 98.25 | 60.75 | 91.16 | 95.65 | 94.19 |
| 3B - Named | 94.00 | 69.50 | 94.11 | 97.11 | 95.06 |
| 3B - Positional | 99.00 | 72.00 | 93.92 | 97.42 | 95.62 |
| 7B - Named | 96.5 0 | 77.50 | 95.01 | 96.96 | 97.65 |
| 7B - Positional | 99.00 | 79.25 | 94.81 | 96.58 | 98.43 |
| Qwen-2.5-Math-Instruct | | | | | |
| 1.5B - Named | 97.50 | 59.75 | 91.68 | 95.4 | 94.08 |
| 1.5B - Positional | 97.25 | 53.25 | 90.19 | 94.69 | 94.18 |
| 7B - Named | 96.75 | 70.25 | 93.40 | 96.45 | 96.26 |
| 7B - Positional | 99.50 | 68.50 | 92.42 | 95.94 | 94.86 |

Table 9: Performance (%) of results delivered by the ChatGPT API using gpt-4.1-mini and gpt-4.1.

| Output type | Compile (%) | Pass (%) | Instance F1 (%) | Argument F1 (%) | IoU (%) |
|---|---|---|---|---|---|
| ChatGPT (gpt-4.1-mini) | | | | | |
| Code w/ named argument | 90.00 | 6.25 | 68.89 | 83.62 | 88.40 |
| Code w/ positional argument | 93.75 | 3.75 | 68.33 | 84.37 | 94.80 |
| Coordinate | - | - | - | - | 27.06 |
| ChatGPT (gpt-4.1) | | | | | |
| Code w/ named argument | 97.50 | 22.50 | 78.54 | 90.49 | 90.03 |
| Code w/ positional argument | 93.75 | 23.75 | 76.67 | 88.52 | 88.95 |
| Coordinate | - | - | - | - | 29.94 |

series, while exhibited strong performance in mathematical reasoning tasks, lagged behind in this fine-grained, specialized code generation context. Furthermore, for all model types, performance generally improved with larger model scales, highlighting the advantage of increased capacity.

Second, when considering argument type effect, positional arguments yielded higher compile rate across models, likely due to their shorter token lengths and syntactic simplicity. However, for other metrics, both argument types demonstrate comparable performance.

# F   Performance of proprietary models

Proprietary large language models have become increasingly accessible and are often deployed as co-pilots for practical tasks. To assess their applicability, we evaluated two representative proprietary models, i.e., gpt-4.1-mini and gpt-4.1, on the MBL generation task, using code and coordinates as outputs, respectively. The specific prompts used for this evaluation can refer to Appendix J.

Table 10: Evaluation on combinations of various numbers of modules, units, and rooms. Case counts are indicated in parentheses. Qwen2.5-Coder-3B was fine-tuned to output code with positional arguments.

| Number | Pass (%) | Instance F1 (%) | Argument F1 (%) | IoU (%) | Module IoU (%) | Unit IoU (%) | Room IoU (%) |
|---|---|---|---|---|---|---|---|
| Module | | | | | | | |
| All (80) | 72.00 | 93.92 | 97.42 | 95.62 | 97.69 | 96.09 | 93.56 |
| ≤2 (38) | 79.47 | 92.72 | 96.49 | 95.79 | 99.43 | 96.23 | 93.36 |
| >2 (42) | 65.24 | 94.61 | 97.95 | 95.56 | 96.57 | 96.14 | 93.95 |
| Unit | | | | | | | |
| All (80) | 72.00 | 93.92 | 97.42 | 95.62 | 97.69 | 96.09 | 93.56 |
| 1 (64) | 77.50 | 96.61 | 98.73 | 97.22 | 97.37 | 97.29 | 97.00 |
| 2 (16) | 50.00 | 85.16 | 93.02 | 92.33 | 98.64 | 93.66 | 88.07 |
| Room | | | | | | | |
| All (80) | 72.00 | 93.92 | 97.42 | 95.62 | 97.69 | 96.09 | 93.56 |
| ≤4 (32) | 79.38 | 90.58 | 94.98 | 94.10 | 99.26 | 94.58 | 90.47 |
| >4 (48) | 67.08 | 95.17 | 98.29 | 97.01 | 97.01 | 96.84 | 95.25 |

Table 11: Evaluation on combinations of various numbers of modules, units, and rooms. Case counts are indicated in parentheses. Qwen2.5-Coder-3B was fine-tuned to output coordinate sequences.

| Number | IoU (%) | Module IoU (%) | Unit IoU (%) | Room IoU (%) |
|---|---|---|---|---|
| Module | | | | |
| All (80) | 81.94 | 87.87 | 87.99 | 72.63 |
| ≤2 (38) | 87.51 | 99.66 | 97.85 | 75.32 |
| >2 (42) | 78.21 | 81.84 | 82.67 | 70.07 |
| Unit | | | | |
| All (80) | 81.94 | 87.87 | 87.99 | 72.63 |
| 1 (64) | 83.84 | 85.65 | 87.06 | 78.88 |
| 2 (16) | 78.03 | 94.88 | 90.19 | 63.78 |
| Room | | | | |
| All (80) | 81.94 | 87.87 | 87.99 | 72.63 |
| ≤4 (32) | 87.11 | 99.54 | 97.33 | 74.44 |
| >4 (48) | 79.98 | 84.02 | 85.11 | 71.84 |

As shown in Table 9, although the model rarely produces fully correct code, it demonstrates a strong ability to extract accurate information from unstructured text inputs, particularly with respect to geometric details. Notably, the generation of coordinate sequences remains a significant challenge, suggesting that precise spatial reasoning continues to be a limitation for current proprietary models in design-related domains.

## G   Effect of the number of modules, units, and rooms

In this section, we group MBL designs based on the numbers of constituent components, i.e., modules, units, and rooms, to assess the impact of component numbers. More components generally correspond to more complicated spatial arrangement, thus raising task difficulty. Each group was formed to ensure both comparative diversity and sufficient sample size.

Table 10 displays the performance when code is used as the output format. As the number of components increased, we observe a general decline in pass rates, reflecting the heightened difficulty of generating syntactically and semantically correct code in more complex MBL scenarios. For modules and rooms, other evaluation metrics remained relatively stable or even exhibited improvements. This suggests that the designed code architecture effectively captures MBL design principles, reducing the

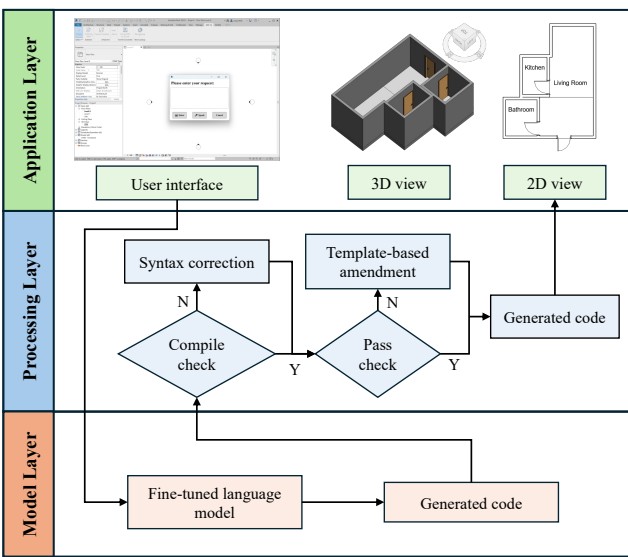

Figure 8: The flow diagram of applying Text2MBL in BIM.

burden on models to reason about each component in isolation. These results highlight the strength of the Text2MBL framework in shifting the challenge from low-level spatial reasoning to high-level semantic understanding.

The performance trend observed for units, however, is less straightforward. Since the unit is an intermediate concept between module and room, and the maximum number of units in the test set is limited to two, it does not reliably indicate overall layout complexity. In fact, designs with one unit contain an average of 2.8 modules and 4.3 rooms, whereas those with two units include 2.3 modules and 5.6 rooms. Consequently, Module IoU is lower for designs with one unit, while Room IoU is higher, reflecting the subtle trade-offs in layout composition.

Similarly, Table 11 reports the results with coordinate sequences as the output format. In contrast to the code-based representation, performance dropped more sharply as the number of modules and rooms increased. This is expected, as coordinate-based outputs relied on independently inferring component coordinates, making it harder for models to generalize across complex scenarios. Results for different unit groupings were consistent with those observed in Table 10. The steeper performance drop compared with code output further validates the superiority of the proposed Text2MBL framework.

# H  Application and deployment

The application and deployment of the proposed Text2MBL are depicted in Fig. 8, which comprises three interconnected layers: the model layer, the processing layer, and the application layer.

At the top, the application layer provides a user interface through which design requirements can be expressed in natural language. These requirements are ultimately translated into BIM-based MBLs, enabling direct user interaction with the design process.

At the foundation lies the model layer, responsible for fine-tuning language models to produce domain-specific code. This code is designed to be executable within BIM environments, facilitating the automated generation of initial MBLs in the early design phase in the construction workflow. However, as discussed in Section 4.3, the raw code generated by the model may occasionally contain errors that impede successful rendering.

To address this, a processing layer is introduced between the model and application layers, functioning as a post-generation refinement stage. This layer ensures compatibility and correctness through a two-step validation process: a compile check and a pass check, both grounded in the code architecture formalized in Section B. Each class and function in the generated code adheres to a predefined schema, allowing automated detection and correction of syntax-level errors during the compile check.

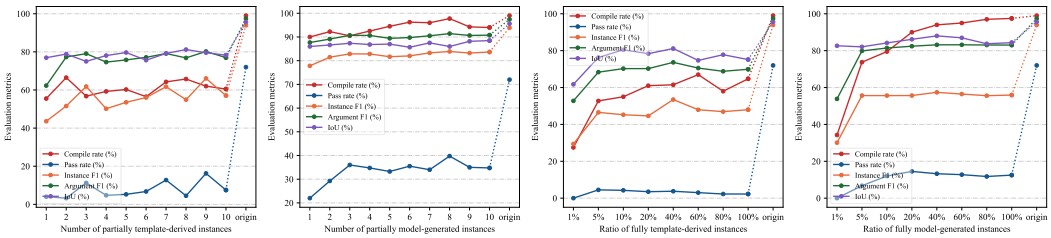

Figure 9: Application of synthetic data with varying numbers (in the partially synthetic setting) and ratios (in the fully synthetic setting), using Qwen2.5-Coder-3B with positional arguments.

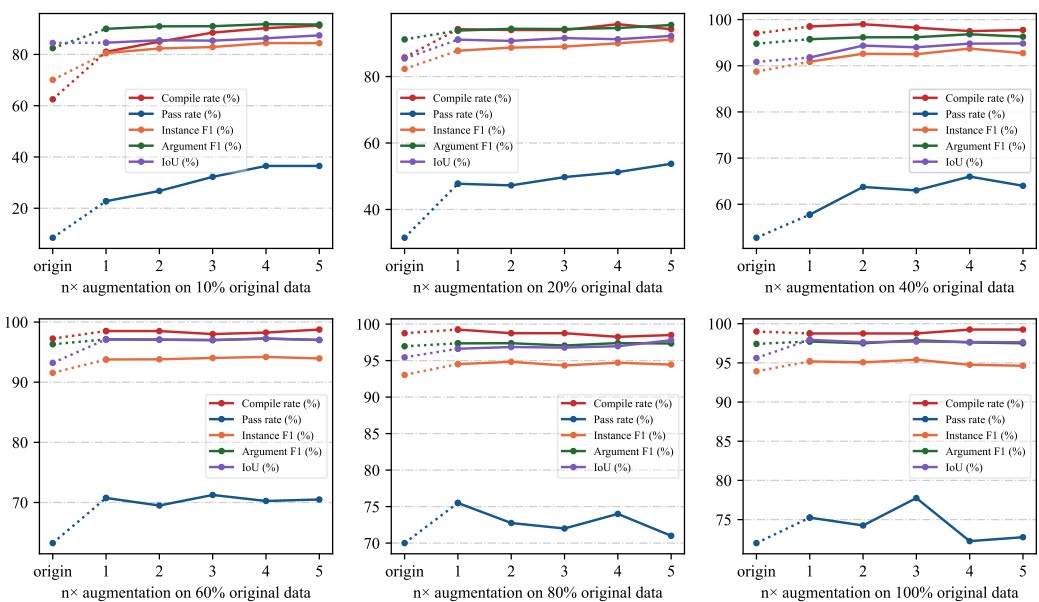

Figure 10: Performance with n× augmentation with partially synthetic template-derived data on $x\%$ of the original data, using Qwen2.5-Coder-3B with positional arguments.

For the pass check, templates and default parameter values are leveraged to repair semantically incomplete or incorrect code segments, ensuring that the final output is executable. The feasibility is demonstrated by strong compile rate and information extraction metrics. The compile rate reflects the system's ability to successfully execute code and generate BIM models. Minor errors that arise during instruction interpretation can often be resolved through compile check, supporting iterative refinement. Meanwhile, information extraction metrics evaluate the system's capability to accurately identify key parameters and capture the intended structure of MBLs within BIM. High performance on both metrics suggests that the system is well-suited for initial deployment in practical settings.

After processing, the refined code is executed and deployed in BIM environments to construct MBLs, which users can further review and customize as needed.

# I Performance of synthetic generated data and augmentation scale

To enrich the training data for MBL generation, we adopt a synthetic data generation strategy. In the partially synthetic setting, we augmented the original training set by generating ten textual descriptions for each available code sequence. In the fully synthetic setting, we constructed code sequences from scratch, which are then converted into corresponding textual descriptions. Both settings were implemented with template-based approach (by using predefined templates to constitute

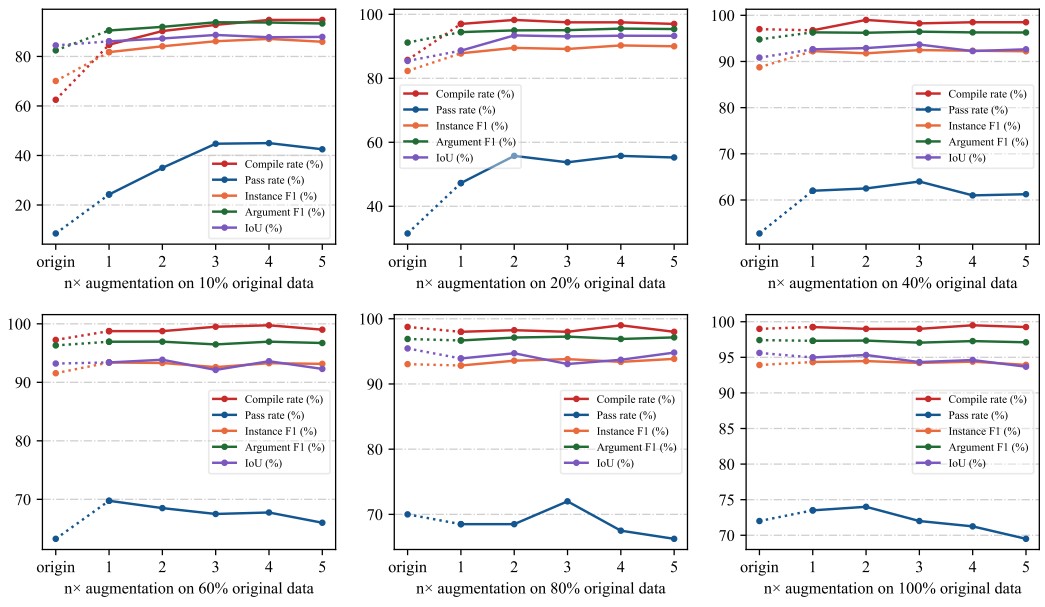

Figure 11: Performance with n× augmentation with partially synthetic model-generated data on $x\%$ of the original data, using Qwen2.5-Coder-3B with positional arguments.

textual descriptions) and model-based approach (by utilizing ChatGPT via gpt-4.1-mini to produce natural language descriptions).

We first evaluated on the ground truth test set using solely synthetic training data. As shown in Fig. 9, training exclusively on synthetic data led to suboptimal performance. Although the information extraction ability exhibited relatively strong, the pass rate, which directly reflects code generation quality, was notably low. Among the synthetic strategies, the model-based approach yielded higher performance than the template-based one, due to its more fluent and diverse language. However, the overall performance remains unsatisfactory. We attribute this discrepancy to the rigid and repetitive description logic of synthetic descriptions, limiting models ability to generalize beyond seen patterns. Despite being constructed entirely from fabricated code, the fully synthetic dataset achieved performance comparable to that of the partially synthetic counterpart. This result suggests that well-designed synthetic data, even when entirely artificial, can support model training to some extent.

We then assess the effect of different augmentation scales using synthetic data under partially synthetic and fully synthetic settings using template- and model-based approaches. The results of utilizing partially synthetic template-derived data, partially synthetic model-generated data, fully synthetic template-derived data, and fully synthetic model-generated data are presented in Fig. 10, Fig. 11, Fig. 12, Fig. 13, respectively.

These four tables reveal that, in low-resource scenarios, incorporating synthetic data markedly boosted model performance across all evaluation metrics, highlighting the utility of synthetic data as a valuable supplement when real-world training data is limited in MBL design contexts. As the proportion of original data increased, most metrics (i.e., compile rate, instance F1, argument F1, and IoU) remained stable in a high level. However, the pass rate lagged behind other metrics and obtained continuous improvements with synthetic data, implying that it is a more challenging criterion and may require more sophisticated strategies for further improvement. Moreover, when the proportion of original data reached moderate to high levels (e.g., 60% to 100%), further scaling of synthetic data (e.g., with scale factors of 4 or 5) yields diminishing or even negative returns. This saturation effect suggests that excessive synthetic augmentation may introduce redundancy or noise, which can undermine the benefits of additional training data.

# J Prompts used

In this section, we provide the prompts utilized in our study.

## J.1 Prompts used for the direct use of proprietary models

To fully leverage the capabilities of proprietary models (i.e., gpt-4.1-mini in our implementation), we designed structured prompts tailored to each task. Each prompt begins with a high-level task description to guide the model's understanding, followed by task-specific contextual information (e.g., core classes and functions within the code architecture). We then specify detailed task requirements and include illustrative examples to further constrain the model's output space and promote consistency. The actual input is appended at the end of the prompt to trigger the desired generation behavior.

Figure 14, 15, and 16 depict the prompt templates used for three tasks: synthetic data generation (i.e., translating source code into textual descriptions), code generation (i.e., producing BIM code from textual descriptions), and coordinate generation (i.e., inferring spatial coordinate sequences from textual descriptions), respectively. During synthetic data generation, the temperature was set to 0.5 to encourage output diversity. For code and coordinate generation, the temperature was set to 0 for deterministic decoding.

## J.2 Prompts used for fine-tuning large language models

During fine-tuning, we employed a straightforward instruction prompt to guide the model: *Please transform the given description into a code-implemented modular building layout*.

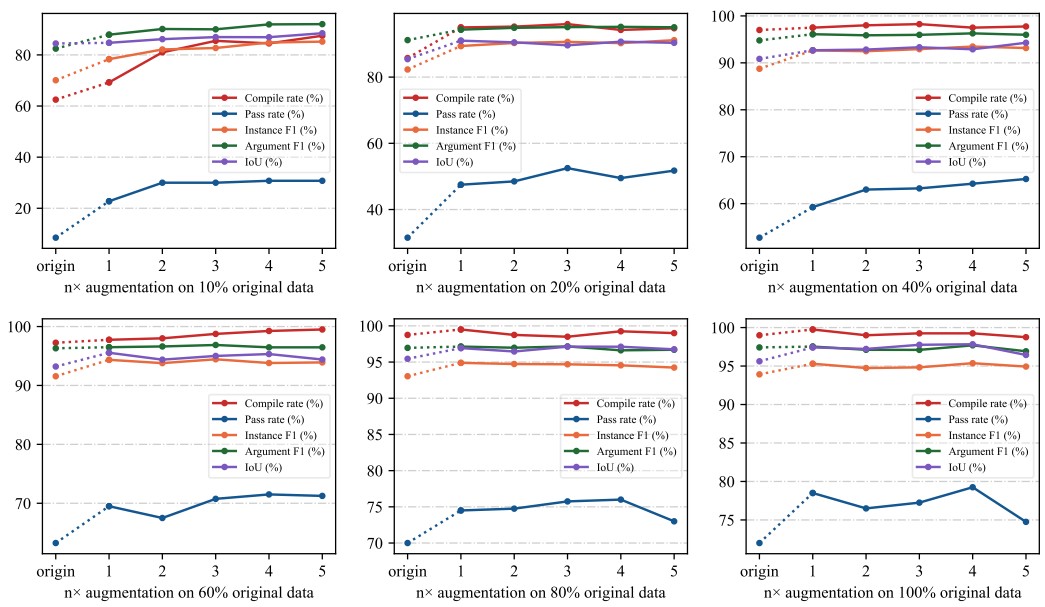

Figure 12: Performance with n× augmentation with fully synthetic template-derived data on $x\%$ of the original data, using Qwen2.5-Coder-3B with positional arguments.

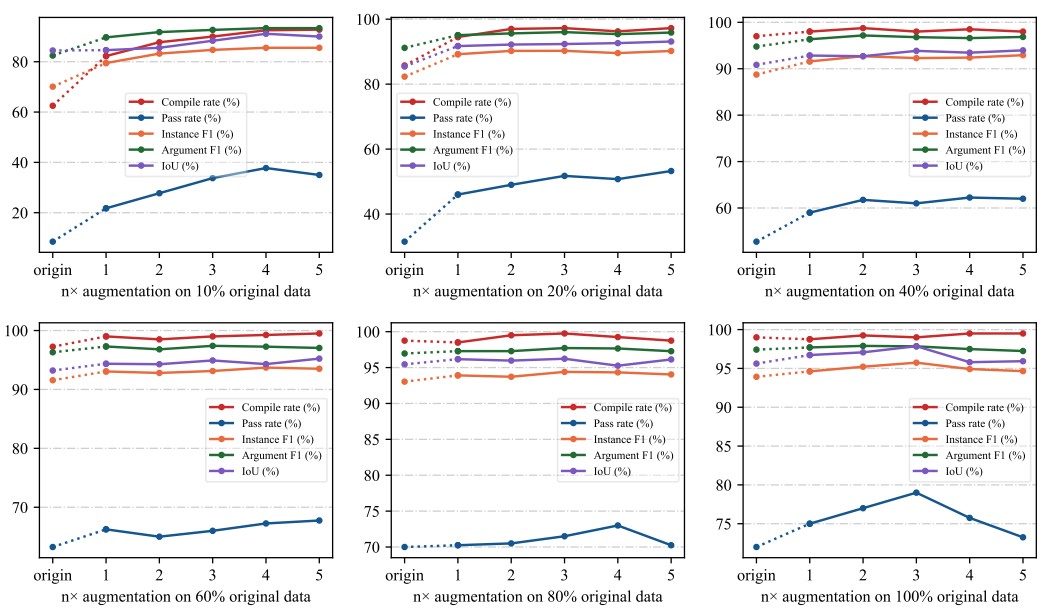

Figure 13: Performance with n× augmentation with fully synthetic model-generated data on $x\%$ of the original data, using Qwen2.5-Coder-3B with positional arguments.

# Overview
You are tasked with generating textual descriptions for Modular Building Layout (MBL) designs derived from C#-based Revit API code. Acting as a user interface layer, your role is to interpret the executable code and produce clear, structured, and semantically accurate descriptions.

# Core Classes and Functions

## 1. Module Class
Represents a fundamental physical cell in an MBL design within a Revit document.

**Constructors:**
```csharp
Introduction to constructors
```

## 2. Unit Class
Organizes collections of modules and rooms within a Revit document, representing a self-contained dwelling space in an MBL design.

**Constructors:**
```csharp
Introduction to constructors
```

## 3. Room Class
Represents functional spaces in modules or units within a Revit document.

**Constructors:**
```csharp
Introduction to constructors
```

## 4. Utils Class
Provides advanced operations for modifying and enhancing MBL designs.

**Functions:**
```csharp
Introduction to functions
```

# Task Requirements

1. Analyze the provided C# code carefully and generate corresponding textual descriptions
2. You may rearrange the description logic, but be sure to maintain the hierarchical containment constraints: each unit must be fully enclosed within a union of one or more modules and each room must be nested within a specific unit
3. During generation, try to employ diverse vocabulary and syntactic structures to enhance variation
4. Only output the textual description; do not include any explanations, comments, or additional text

# Examples Implementations
*Three demonstrations*

# Input Code
*Input*

Figure 14: prompt used for using the proprietary model in synthetic data generation.

**Prompt for code generation**

# Overview
You are tasked with generating C# code for Modular Building Layout (MBL) design in Revit based on textual descriptions. As a specialist with expertise in both architectural modular design and Revit API programming, you will translate textual descriptions into executable code using the provided framework.

# Core Classes and Functions

## 1. Module Class
Represents a fundamental physical cell in an MBL design within a Revit document.

**Constructors:**
```csharp
Introduction to constructors
```

## 2. Unit Class
Organizes collections of modules and rooms within a Revit document, representing a self-contained dwelling space in an MBL design.

**Constructors:**
```csharp
Introduction to constructors
```

## 3. Room Class
Represents functional spaces in modules or units within a Revit document.

**Constructors:**
```csharp
Introduction to constructors
```

## 4. Utils Class
Provides advanced operations for modifying and enhancing MBL designs.

**Functions:**
```csharp
Introduction to functions
```

# Task Requirements

1. Analyze the provided textual description of a building layout
2. Generate valid C# code implementing the described MBL design using the framework above
3. Ensure code correctness and adherence to the provided API
4. Follow a systematic approach similar to the provided examples
5. Only output the C# code; do not include any explanations, comments, or additional text

# Examples Implementations
*Three demonstrations*

# Input Textual Description
*Input*

Figure 15: prompt used for using the proprietary model in code generation.

**Prompt for coordinate generation**

# Overview
You are tasked with generating precise spatial coordinates of components (i.e., modules, units, and rooms) for Modular Building Layout (MBL) designs based on textual descriptions. As a specialist with expertise in architectural modular design, you will translate textual descriptions into coordinate sequences.

# Task Requirements

1. Analyze the provided textual description of a building layout
2. Generate coordinates of each component (i.e., module, unit, and room) for the description
3. Follow a systematic approach similar to the provided examples
4. Only output the coordinate sequence; do not include any explanations, comments, or additional text

# Examples Implementations
*Three demonstrations*

# Input Textual Description
*Input*

Figure 16: prompt used for using the proprietary model coordinate generation.

