# OpenReview forum: "Text-to-Code Generation for Modular Building Layouts in Building Information Modeling"
_NeurIPS.cc/2025/Conference — NeurIPS 2025 poster_

### Official Review · Reviewer_UQki · 2025-06-11

**Clarity:** 3
**Significance:** 3
**Originality:** 3
**Rating:** 4
**Confidence:** 4

**Summary:**

This paper proposes a new method for generating BIM models in a text-guided manner by representing BIM as executable code and framing the generation task as a sequence-to-sequence problem. The main technical contributions include:
-	A code representation scheme specifically designed based on Modular Building Layout (MBL) principles.
-	A newly curated dataset containing paired textual descriptions and corresponding structured BIM code.

The authors fine-tuned a large language model using the proposed dataset to generate structured BIM code, and evaluated the performance using metrics related to code validity, semantic fidelity, and geometric consistency. The experimental results demonstrate the effectiveness of the proposed code representation in enabling autonomous BIM generation.

**Questions:**

1. It would be helpful to understand how the large language model performs in a zero-shot setting. Specifically, could the authors report the performance of the pre-trained model (without fine-tuning) under both the proposed code representation and the coordinate-based representation? Such a comparison would not only help validate whether the proposed code representation is more suitable for the task but also provide insights into the effectiveness of the proposed dataset.

2. What is the generalization capability of the fine-tuned model? Since the model is trained on a specific dataset, there is a risk that it may overfit to the style or phrasing patterns in the training examples. The reviewer would like to know how well the model performs when given free-form, less structured textual inputs — for instance, a broad requirement such as: "I need a house with a living room, three bedrooms, a bathroom, a kitchen, and a balcony."

**Ethical Concerns:**

["NO or VERY MINOR ethics concerns only"]

**Limitations:**

yes

**Paper Formatting Concerns:**

No formatting concerns were identified.

**Quality:**

3

**Strengths And Weaknesses:**

Strengths:
- The paper is well-written and clearly presented.
- The technical contribution centers on the design of a code representation tailored for BIM generation, which aligns well with MBL modeling principles. In addition, the authors have curated a new dataset consisting of paired textual descriptions and corresponding executable BIM codes, which is an important resource that supports future research in this area.
- The experimental evaluation is relatively comprehensive, covering multiple aspects such as code validity, semantic fidelity, and geometric consistency.

Weakness:
- The model architecture lacks significant innovation. The approach mainly relies on fine-tuning an existing large language model using a newly constructed dataset, without introducing substantial modifications to the underlying architecture or training methodology. As a result, the technical novelty of the work may be limited. Additionally, the reviewer would like to raise a few concerns about the performance of the large language model; see “questions” for more details.

---

> ### Author Rebuttal · Authors · 2025-07-31
>
> We would like to express our sincere gratitude to the reviewer for the thoughtful and constructive comments. After a thorough analysis of the reviewer’s comments, we have synthesized the key points into three areas: (1) framework novelty, (2) zero-shot setting evaluation, and (3) generalization capability.
>
> To facilitate review, we have prepared a detailed response to each point raised.
>
> **1 Framework novelty (W1):**
>
> We regret not having clearly stated our motivation and contributions in our manuscript. We appreciate the opportunity to clarify the motivation and contributions of our work. We acknowledge that our approach does not propose architectural innovations or new training methodologies for LLMs. Instead, our focus is on addressing the unique challenges of MBL generation within BIM environments, a domain characterized by complex model structures and intricate design constraints.
>
> Recognizing that progress in this area depends not only on model architecture but also on the quality and specificity of the data and task formulation, we undertook a comprehensive review of existing MBL designs and systematically distilled key design principles relevant to text-based generation. Building upon this foundation, we carefully constructed a new dataset and developed a code architecture to represent MBLs in BIM environments using modular classes and functions. This enables the encoding of high-level commands implemented with detailed operational logic, facilitating a new paradigm for automating MBL generation in this context.
>
> It is worth mentioning that our proposed framework introduces an efficient and expressive model representation for BIM-based MBLs by encoding design intent and structural information as action sequences in a code format. While our experimental comparisons include coordinate-based approaches, such representations are inherently limited in capturing complex design requirements—such as hierarchical relationships, component sequencing, non-rectilinear geometries, and rich semantic associations—essential in practical MBL applications. Therefore, it is inapplicable to represent BIM models with coordinates in design-construction workflows in real-world settings. The Industry Foundation Classes (IFC) standard is widely adopted for BIM interoperability to serialize BIM models; however, its verbose and semantically redundant structure leads to substantial file sizes and high token counts, posing challenges for downstream tasks, particularly those involving natural language processing or machine learning. In contrast, our code architecture compresses BIM models into succinct, encapsulated classes and functions, dramatically reducing both the representation size and complexity. For instance, an MBL comprising five modules, one unit, and six rooms typically requires 7503 lines (215607 tokens) in IFC format, whereas our approach achieves equivalent expressivity with only 20 lines (884 tokens). This compact and structured representation not only enhances computational efficiency but also facilitates more effective learning and reasoning in data-driven applications.
>
> While our work leverages established large language models, we believe our contributions lie in integrating domain-specific knowledge with LLM capabilities, formulating a novel code architecture for BIM-based MBLs, creating a corresponding dataset, and demonstrating practical workflows for BIM-based MBL generation. We have conducted extensive experiments to validate the effectiveness of this framework. We maintain that while powerful backbone models are constantly evolving, the fundamental challenge lies in capturing the core of the problem and developing succinct yet effective solutions to address domain-specific scenarios.
>
> In response to this useful comment, we will revise our Introduction, Experiment, and Discussion sections to more explicitly articulate our motivation, clarify the scope of our contributions, and discuss the significance of our approach in the broader context of BIM and automated code generation.
>
> **2 Zero-shot setting evaluation (Q1):**
>
> We sincerely appreciate the reviewer’s thoughtful suggestion regarding the evaluation of LLMs in the zero-shot setting. We agree that reporting the performance of pre-trained models—without any fine-tuning—on both our proposed code representation and the conventional coordinate-based representation would provide a clearer assessment of the suitability of these approaches, as well as further insights into the effectiveness of the proposed dataset.
>
> In Figure 6, we have shown smaller fine-tuned models in low-resource scenarios. Our additional experiments used several state-of-the-art, general-purpose LLMs, including GPT-4.1, GPT-4.1-mini, Gemini-2.5-pro, and Gemini-2.5-flash, in the zero-shot scenario. For each model, we evaluated performance on the proposed code representations (using named arguments or positional arguments) as well as on the coordinate-based representation, using the metrics introduced in the main paper. The results are summarized in the following table:
>
> |Model and output|Compile|Pass|Instance F1|Argument F1|IoU|
> |-|-|-|-|-|-|
> |Gpt-4.1-mini||||||
> |Name|90|6.25|68.89|83.62|88.4|
> |Potision|93.75|3.75|68.33|84.37|94.8|
> |Coordinate|-|-|-|-|27.06|
> |Gpt-4.1||||||
> |Name|97.5|22.5|78.54|90.49|90.03|
> |Potision|93.75|23.75|76.67|88.52|88.95|
> |Coordinate|-|-|-|-|29.94|
> |Gemini-2.5-flash||||||
> |Name|95|21.25|73.58|83.49|84.81|
> |Potision|92.5|7.5|65.13|75.61|88.47|
> |Coordinate|-|-|-|-|23.94|
> |Gemini-2.5-pro||||||
> |Name|97.5|36.25|84.18|88.57|90.24|
> |Potision|98.75|35|84.95|92.02|93.86|
> |Coordinate|-|-|-|-|30.72|
>
> These results indicate that, in the zero-shot setting, though it is sill challenging for modern LLMs to output totally accurate code, they exhibit a strong capacity to extract and structure information from unstructured textual descriptions, especially when using our proposed code representations. However, generating precise coordinate sequences remains a significant challenge for all evaluated models, with performance of IoU on the coordinate-based representation substantially lower than that for code representations. This suggests that spatial reasoning based on raw coordinates is still a major limitation for current LLMs in this context. Moreover, conventional approaches still fall short in generating MBLs within BIM environments with multiple constraints. In contrast, our proposed framework can dynamically generate the corresponding code for on-the-fly BIM model installation.
>
> These findings reinforce the value of our code-based formulation and indicate that it is more tractable for LLMs, especially in zero-shot scenarios. Additionally, the results suggest that our dataset provides a challenging and informative benchmark for future model development.
>
> We acknowledge this valuable feedback, and we will update the Experiment section and the corresponding appendix to incorporate these new results and further clarify their implications.
>
> **3 Generalization capability (Q2):**
>
> We sincerely appreciate the reviewer’s insights, which are truly enlightening and invigorating. We agree that evaluating the fine-tuned model’s performance on free-form and less structured inputs is crucial to understanding its broader applicability, generalization capability, and potential limitations. Our current work is primarily focused on parametric design scenarios that utilize detailed and structured inputs, rather than general, configurator-style user requirements. However, we recognize the importance of addressing a wider range of input granularities, and this suggestion highlights an important direction for our future research.
>
> To address this valuable concern, we have conducted additional experiments in which the models were provided with more abstract and less structured instructions (e.g., “Generate a layout with x module, x unit, x living room, x bathroom, x bedroom, x kitchen”). We evaluated the models on two metrics: compile rate (i.e., successful generation of executable code) and extraction F1 score (measuring the accurate extraction and mention of modules, units, and rooms). The results are summarized below:
>
> |Model and output|Compile|F1|
> |-|-|-|
> |Qwen2.5-Coder-3B|||
> |Name|68.75|99.05|
> |Position|45|98.36|
> |Qwen2.5-Coder-7B|||
> |Name|92.5|100|
> |Position|95|99.53|
> |Gpt-4.1-mini|||
> |Name|81.25|92.91|
> |Position|78.75|91.96|
> |Gpt-4.1|||
> |Name|87.5|96.11|
> |Position|82.5|94.58|
>
> These results indicate that the models exhibit a basic level of generalization to free-form and less structured queries, though performance varies by model size. In particular, larger models tend to generate more accurate and executable outputs, while smaller models are more prone to errors in code generation. This suggests that while our approach can handle a degree of abstraction, further improvements are necessary to robustly support the full granularities of user input styles.
>
> We are grateful for the feedback, which has motivated us to both broaden our experimental evaluation and to consider a more comprehensive discussion of generalization in the manuscript. We will update the Experiment and Discussion sections accordingly to reflect these new findings and to clarify the scope and limitations of our current approach.

---

> > ### Comment · Reviewer_UQki · 2025-08-06
> >
> > My initial concerns centered on three main issues. First, the proposed framework does not introduce architectural innovations or novel training techniques for large language models. Second, I suggested including a comparison of the zero-shot performance of the pre-trained LLM under both the proposed code-based representation and the traditional coordinate-based format. Third, I raised the question of how well the fine-tuned LLM generalizes beyond the specific training scenarios.
> >
> > Regarding the first point, the authors clarified that their primary technical contribution lies not in altering the LLM architecture but in the proposed code representation. By encapsulating design intent and structural relationships within modular classes and functions, their approach significantly reduces complexity relative to the verbose IFC format and better captures semantic constraints than raw coordinate sequences. The authors’ reasonable explanation resolves my first concern
> >
> > As for the second concern, the authors present zero-shot results for several SOTA LLMs. Although none of the models output totally accurate code, as evidenced by relatively low pass rates, the proposed representation consistently yields substantially higher IoU scores compared to the coordinate-based baseline. These results support the claim that their code representation is more compatible with the current capabilities of LLMs. I find that these experiments adequately address my request.
> >
> > Finally, the authors added experiments using abstract, free-form instructions to evaluate generalization. As expected, model performance declines compared to detailed prompts, yet even smaller models show a reasonable ability to handle abstraction after fine-tuning on the proposed dataset. Given that the models were fine-tuned specifically to this domain, I consider the generalization findings both credible and informative.
> >
> > In conclusion, the authors’ detailed rebuttal and additional experiments have addressed all of my concerns. I therefore stand by my original positive recommendation.

---

> > > ### Author Response · Authors · 2025-08-09
> > >
> > > We sincerely thank the reviewer for the thorough and thoughtful engagement throughout the review process. We deeply appreciate the time and effort the reviewer devoted to understanding our work and providing constructive feedback. The dedication to the review process has been instrumental in strengthening our paper.

---

### Official Review · Reviewer_ZzJA · 2025-07-03

**Clarity:** 3
**Significance:** 3
**Originality:** 2
**Rating:** 5
**Confidence:** 3

**Summary:**

Motivated by applications in user-inclusive design in construction workflow, this paper presents Text2MBL, a text-to-code generation framework that takes in textual description and generates executable Building Information Modeling (BIM) code for modular building layout (MBL) design.

To account for MBLs’ hierarchical three-tier structure (module, unit, and room), the authors design a specific object-oriented code structure and fine-tune LLMs to generate code of this structure. Specifically, the output takes the form of action sequences that when execuetd in BIM environment produce MBLs.

To fine-tune the LLM, data is curated from real-world housing projects, and synthetic data is shown to enhance peformance significantly when used for augmentation.

Text2MBL is shown to outperform conventional coordinate-based approaches in geometric consistency, and perform strongly on executable validity, semantic fidelity.

**Questions:**

- The only baseline is coordinate-based approach. Is this predominantly limited by the fact that few existing methods output BIM-based MBLs? Is it entirely unreasonable to compare with other text-to-floorplan methods (in aspects such as no missing entity, correct inter-entity relation, geometric consistency IoU, etc.)?
- It is mentioned that critical factors such as element placement, clash avoidance, component order, and support for irregular geometries, are addressed by Text2MBL by “abstracting complex design operations into high-level interfaces” (line 223-224). What exactly does this mean?
- The relationship between BIM and MBL is not explicitly explained. How does “BIM model” relate to “MBL” (line 102-103)? What is different about “BIM-based MBL” compared to standard MBL?  Also, for someone not familiar with the specific application domain, the use of “BIM” (without definition up front) in the introduction can be a source of confusion.

**Ethical Concerns:**

["NO or VERY MINOR ethics concerns only"]

**Final Justification:**

I appreciate the authors’ thorough response to my questions and concerns and will maintain my rating of ‘accept’.

**Limitations:**

Yes.

**Quality:**

3

**Strengths And Weaknesses:**

Quality
- [Strength] The method used (synthetic data augmentation, code format, LLM finetuning)  is not particularly complex but is well grounded in the application and well supported by experimental result.
- [Strength] Result analysis is comprehensive.
- [Strength] There are multiple discussions of limitations.

Clarity
- [Strength] The paper is generally well written, and a comprehensive appendix is provided.

Significance
- [Strength] This work is clearly well grounded in application, and practitioners can likely benefit from it.
- [Strength] Code and data are both released, and a functional plug-in is deployed on a representative BIM platform. Deployed as detailed in Appendix H.
- [Weakness] The current model is not particularly expressive, which can be a topic of future work.

Originality
- [Strength] The specific task, generating BIM-based MBL from text is underexplored, and this work proposes a new method that outperforms existing method.
- [Weakness] There are lines of works in the vision community that tackeles the general text-to-scene task that are not mentioned as related works. For example, “AnyHome” generates house floorplan from text and eventually a detailed mesh, and “LLplace” and “Open-Universe Indoor Scene Generation using LLM Program Synthesis and Uncurated Object Databases” generate code or structured program for 1-room scene. Constraints for modular construction can likely be inserted by fine-tuning the LLM or modifying the prompt.
- [Weakness] The high-level insight that code-driven approach is effective has been discussed in various works in the LLM community.

---

> ### Author Rebuttal · Authors · 2025-07-31
>
> We would like to thank the reviewer for the insightful and constructive feedback. We have carefully extracted and categorized the valuable concerns into the following three areas: (1) concept confusion, (2) method comparison and performance, and (3) discussion on LLM-based design tools.
>
> For ease of review, we have structured our response to correspond with each area, including thorough revision notes where applicable.
>
> **1 Concept confusion (Q2 and Q3):**
>
> We recognize that the introduction of “BIM” and “BIM-based MBL” without a proper definition may confuse, especially for readers less familiar with the domain. We appreciate the opportunity to elaborate on these foundational aspects.
>
> BIM is a digital process for creating and managing rich, structured representations of a building’s physical and functional characteristics. Unlike traditional CAD drawings, which only capture geometric information, BIM integrates geometry, spatial relationships, material properties, and construction metadata into a parametric model. For example, in a BIM model, a “wall” is not just a line but an object with height, thickness, material type, thermal performance, and construction sequence attributes. BIM platforms allow designers to collaboratively develop models that are used across the lifecycle of a building—for design coordination, clash detection, scheduling, cost estimation, and facility management.
>
> We focus on BIM due to its importance in industrialized construction, which emphasizes prefabrication and integrates manufacturing expertise. Modular construction represents a masterful synthesis, utilizing factory-made modules that are transported and assembled onsite. Consequently, lifecycle management for MBLs becomes more critical. BIM is undoubtedly the best choice for representation. Unlike standard MBLs, which may only define spatial configurations abstractly or on 2D plans, BIM-based MBLs are instantiated as parametric objects within BIM, allowing for more downstream applications.
>
> The process of authoring BIM-based MBLs is significantly more complex than producing schematic MBLs. It requires precise element placement, avoidance of geometric clashes, consideration of component order, and support for irregular module geometries from module operations. These challenges are compounded by the hierarchical relationships among modules, units, and rooms. Current coordinate-driven approaches in MBL design are typically insufficient for these tasks, as they do not address the requirements inherent to BIM instantiation.
>
> To address these challenges, Text2MBL abstracts complex design operations into high-level interfaces. Specifically, we have developed an architecture of classes and functions that encapsulate essential operations at a semantic level. By “abstracting complex design operations into high-level interfaces,” we mean that rather than relying on language models to generate low-level, step-by-step manipulations (which we found to be unreliable in our preliminary attempts), we provide a set of robust, reusable abstractions. For example, the operation of merging modules involves a sequence of geometric and semantic updates—removing internal walls, generating new boundaries, and validating against existing elements—which are encapsulated within dedicated functions, and only one interface is exposed. Similarly, the placement of an indented door requires calculations for positioning, wall modifications, and consistency checks, all of which are implemented within our codebase rather than delegated to language models.
>
> We have adopted this approach after observing that LLMs currently struggle to generate accurate and executable code for these specialized BIM operations using BIM APIs, both because of the niche nature of BIM APIs and the need for precise geometric reasoning.
>
> We appreciate the reviewer’s feedback and will revise the manuscript to define BIM and MBL clearly and to elaborate on the motivations for BIM-based MBLs in the Introduction and Background sections.
>
> **2 Method comparison and performance (W1 and Q1):**
>
> We apologize for not clearly articulating the novelty and scope of our problem. Our work, to our knowledge, is the first to research BIM-based MBLs generation from textual instructions. Existing approaches in BIM community focused on generating BIM models using coordinate-based methods, without modeling the hierarchical, geometric, and semantic relationships inherent in MBLs. Most prior floorplan generation methods in vision community output images based on general user instructions, rather than detailed, parametric BIM representations with explicit hierarchical and semantic structures.
>
> We agree with the reviewer that, in principle, it would be informative to compare our approach with other methods using metrics such as entity coverage and inter-entity relational correctness. However, we note that prior works typically target different outputs (e.g., images) and problem formulations, often with more general instructions and less parametric detail. Our focus is on parametric BIM design, where the generated outputs must satisfy strict code constraints and be executable within BIM. Accordingly, our evaluation has emphasized metrics such as code compile and pass rates, as well as information extraction scores for argument accuracy, which directly reflect the practical applicability of the generated BIM-based MBLs.
>
> The reviewer’s insightful comment encouraged us to reflect more deeply. As user instructions grow more abstract, different evaluation metrics and baselines become necessary. This motivates us to explore further. We have added experiments where models were given abstract instructions (e.g., “Generate a layout with x module, x unit, x living room, x bathroom, x bedroom, x kitchen”). We report both the compile rates and extraction F1 scores (measuring the accurate extraction and mention of modules, units, and rooms).
>
> |Model and output|Compile|F1|
> |-|-|-|
> |Qwen2.5-Coder-3B|||
> |Name|68.75|99.05|
> |Position|45|98.36|
> |Qwen2.5-Coder-7B|||
> |Name|92.5|100|
> |Position|95|99.53|
> |Gpt-4.1-mini|||
> |Name|81.25|92.91|
> |Position|78.75|91.96|
> |Gpt-4.1|||
> |Name|87.5|96.11|
> |Position|82.5|94.58|
>
> The new extraction F1 scores directly address entity coverage. While our main baseline remains the coordinate-based approach, as it is currently most relevant for BIM outputs in industry, we recognize the value of broader comparisons, especially as user instructions become more abstract and align closer to previous text-to-floorplan methods.
>
> Regarding model expressiveness, the relatively high compile rates and information extraction metrics suggest that the models are already suitable for initial BIM-based MBL generation in real-world settings. Improving the pass rates and enhancing the ability to handle more complex or general specifications is an important direction for future work.
>
> In response to the thoughtful feedback, we will revise the Experiment section to clarify our definition, evaluation, and include new experiments. We will also update the Discussion section to address the potential for incorporating additional baselines and evaluation metrics as the field evolves, and as user instructions become more diverse and abstract.
>
> **3 Discussion on LLM-based design tools (W2 and W3):**
>
> We appreciate the opportunity to clarify the distinctions from prior research, e.g., for scientific diagrams [1,2] and indoor scenes [3,4,5], which we regretfully did not sufficiently discuss.
>
> While these prior works have made substantial advances in text-to-scene design across various domains, our focus is specifically on architectural and structural design with an emphasis on parametric modeling, construction constraints, and seamless integration with BIM and MBL workflows. In contrast, much of the vision-based text-to-scene generation literature focuses on arranging objects within a single room or general scene configuration, typically relying on object retrieval and placement guided by general and conceptual descriptions, rather than the constraints and principles necessary for architectural practice and downstream construction.
>
> We also acknowledge that the high-level insight of using code-driven approaches for generation has been explored in previous research, such as in scientific diagram synthesis with TikZ [1] and specialized scene description languages for indoor scenes [5]. However, our work distinguishes itself by introducing a domain-specific code representation tailored for BIM environments with full implementation. This representation is designed to encapsulate complex architectural logic and construction requirements, thereby reducing the burden on LLMs to handle intricate reasoning or low-level geometric generation. For instance, operations that may require multiple steps or ad hoc commands in prior scene description languages can be accomplished with a single, semantically meaningful command in our system, maintaining consistency across modules, units, and rooms. Furthermore, the implementation is closely aligned with BIM software, supporting direct interoperability and real-world applicability.
>
> In response to this insightful comment, we will revise the Introduction, Background, and Discussion sections to position our contributions relative to these important works more thoroughly and to clarify the novelty and practical significance of our code representation and BIM integration.
>
> [1] Belouadi et al., AutomaTikZ: Text-Guided Synthesis of Scientific Vector Graphics with TikZ, ICLR 2024
> [2] Zala et al., DiagrammerGPT: Generating Open-Domain, Open-Platform Diagrams via LLM Planning, COLM 2024
> [3] Yang et al., Llplace: The 3d indoor scene layout generation and editing via large language model, 2024
> [4] Fu et al., AnyHome: Open-Vocabulary Generation of Structured and Textured 3D Homes, ECCV 2024
> [5] Aguina-Kang et al., Open-Universe Indoor Scene Generation using LLM Program Synthesis and Uncurated Object Databases, 2024

---

> > ### Comment · Reviewer_ZzJA · 2025-08-07
> >
> > Thank you for the clarification. It’s very helpful!

---

> > > ### Author Response · Authors · 2025-08-09
> > >
> > > We sincerely appreciate the reviewer's careful reading of our rebuttal and the kind acknowledgment. We are glad that our clarifications were helpful. We want to express our sincere gratitude for the reviewer's valuable feedback.

---

### Official Review · Reviewer_9xp5 · 2025-07-03

**Clarity:** 3
**Significance:** 3
**Originality:** 3
**Rating:** 4
**Confidence:** 3

**Summary:**

The authors propose Text2MBL, a text-to-code generation framework that translates natural language descriptions of modular building layouts (MBLs) into executable code for Building Information Modeling (BIM) systems. Addressing the unique challenges posed by MBLs (e.g., hierarchical structures, spatial relationships, and semantic complexity), the authors design a proof-of-concept implementation in Autodesk Revit. They curate a dataset of 198 MBL designs paired with multiple textual descriptions and code sequences, including both real and synthetic examples. Experiments with fine-tuned Qwen2.5 models show that the code-output approach outperforms coordinate-output baselines in terms of compile rate, semantic fidelity (F1 scores), and geometric consistency (IoU). Additional analyses demonstrate the benefit of synthetic data augmentation and the challenges associated with component complexity.

**Questions:**

- As a non-expert in the architecture and construction domains, I found myself wondering how representative the use of Autodesk Revit is for real-world applications. Given that previous works like Text2BIM employed Vectorworks, could the authors elaborate on what BIM tools are most widely adopted in industry? Specifically, what domain-specific languages or software platforms are considered essential or standard in BIM-based workflows?

**Ethical Concerns:**

["NO or VERY MINOR ethics concerns only"]

**Final Justification:**

I have checked the rebuttal and other reviews, and I appreciate the authors' detailed response to my questions. The rebuttal content did not introduce sufficient content to change my evaluation, so I maintain my original rating.

**Limitations:**

No specific concern

**Paper Formatting Concerns:**

No specific concern

**Quality:**

3

**Strengths And Weaknesses:**

**Strengths**

1. **Novel integration of BIM, code generation, and spatial design**: The paper introduces a new task and methodology for text-to-BIM code generation, which bridges natural language understanding and structured building architectural modeling.
2. **Comprehensive system implementation and dataset curation**: The authors provide a full implementation in Autodesk Revit with detailed class designs and a paired dataset of real and synthetic MBLs, facilitating both reproducibility and future research.
3. **Strong empirical results and insightful error analysis**: Across multiple metrics and model scales, the proposed code-driven generation method consistently outperforms coordinate-based baselines. The authors provide fine-grained error analysis and show effective use of synthetic data in low-resource scenarios.

**Weaknesses**

1. **Lack of external validation and user-centric evaluation**: The work would benefit from qualitative assessment by domain experts (e.g., architects, engineers) or user studies to evaluate the practical value of the generated outputs. Table 2 reports very high scores for top models—does this indicate that the benchmark is too easy and already saturated, or that the model is close to being deployable in real-world settings?
2. **Insufficient discussion of related LLM-based design tools**: The paper briefly mentions works like Text2BIM and Tell2Design, but does not provide detailed comparisons. Additionally, recent research on LLM-guided scientific graphics generation, such as *AutomaTikZ* [A] and *DiagrammerGPT* [B], is highly relevant and should be discussed to position the contribution more clearly within the broader landscape of text-to-structured-design systems

[A] Belouadi et al., AutomaTikZ: Text-Guided Synthesis of Scientific Vector Graphics with TikZ, ICLR 2024

[B] Zala et al., DiagrammerGPT: Generating Open-Domain, Open-Platform Diagrams via LLM Planning, COLM 2024

---

> ### Author Rebuttal · Authors · 2025-07-31
>
> We are grateful to the reviewer for the thoughtful and constructive review. We have carefully organized the corresponding concerns into the following three areas: (1) BIM tools and standards, (2) external validation and realistic deployment, and (3) insufficient discussions on related design approaches.
>
> To ensure clarity, we have provided a point-by-point response to the comments, with detailed explanations of the revisions to be made.
>
> **1 BIM tools and standards (Q1):**
>
> We appreciate the opportunity to clarify BIM tools and standards.
>
> The architecture, engineering, and construction (AEC) industry employs a diverse ecosystem of BIM software, including Autodesk Revit and Vectorworks. Among these, Autodesk Revit is widely regarded as the representative software, particularly in medium to large-scale construction projects and in academic research on parametric and generative design workflows. However, with the proliferation of diverse software solutions, ensuring interoperability across these platforms for various stakeholders remains a challenge. To address this, buildingSMART developed the Industry Foundation Classes (IFC) standard, which serves as a platform-agnostic, open specification for representing and exchanging BIM data. IFC enables seamless communication and data exchange between tools, ensuring that a model authored in one software can be interpreted in another.
>
> In this work, we selected Revit as the implementation environment for two reasons. First, Revit provides highly comprehensive and well-documented programmable APIs (primarily in C#), enabling researchers and developers to build robust and fine-grained extensions that operate as first-class citizens within BIM environment. This makes it particularly suitable for prototyping intelligent design agents and automation systems. Second, Revit offers seamless export capabilities to the IFC format, ensuring that any BIM models generated through our system can be readily imported into other IFC-compliant platforms. In this way, although our pipeline is demonstrated within Revit, the resulting models remain compatible with the broader BIM ecosystem.
>
> While previous studies, such as Text2BIM, utilized platforms like Vectorworks, our focus on Revit aligns with industry adoption patterns and enables deeper integration with parametric design tools (e.g., Dynamo) and advanced automation workflows. Moreover, the underlying methodology is conceptually transferable to other BIM tools that support programmable interaction and IFC export.
>
> Additionally, we wish to clarify our design decisions regarding model representation. While IFC is indispensable for interoperability, it is verbose and semantically redundant, resulting in large file sizes and high token counts when parsed for downstream tasks such as natural language processing. Our code architecture significantly compresses BIM models into concise, encapsulated classes and functions. For example, representing an MBL with five modules, one unit, and six rooms requires 7503 lines (215607 tokens) in IFC, but only 20 lines (884 tokens) using our architecture, facilitating more efficient learning and reasoning with much more structured representations.
>
> We acknowledge that we did not sufficiently elaborate on the motivations and backgrounds. In response to this valuable feedback, we will expand the Introduction and Background sections to provide a more comprehensive explanation of BIM tool adoption, standards, and our rationale for selecting Revit.
>
> **2 External validation and realistic deployment (W1):**
>
> Our current study focuses on text-based parametric design systems for BIM-based MBL, emphasizing the accurate parsing of detailed user instructions and the generation of structured code under multi-layered constraints (e.g., hierarchy, topology, semantics). As such, our evaluation has prioritized quantitative metrics to objectively measure the system’s ability to interpret user intent in executable code. Other related studies that employ qualitative evaluation often rely on broader, more conceptual inputs, which can lead to variations in the generated output depending on different potential parameters.
>
> We acknowledge that we did not sufficiently address the generalization of our system to higher-level, more abstract user instructions. In response, we have conducted additional experiments in which the models were prompted with abstract specifications (e.g., “Generate a layout with x module, x unit, x living room, x bathroom, x bedroom, x kitchen”). We report both the compile rates (i.e., successful generation of executable code) and extraction F1 scores (measuring the accurate extraction and mention of modules, units, and rooms).
>
> |Model and output|Compile|F1|
> |-|-|-|
> |Qwen2.5-Coder-3B|||
> |Name|68.75|99.05|
> |Position|45|98.36|
> |Qwen2.5-Coder-7B|||
> |Name|92.5|100|
> |Position|95|99.53|
> |Gpt-4.1-mini|||
> |Name|81.25|92.91|
> |Position|78.75|91.96|
> |Gpt-4.1|||
> |Name|87.5|96.11|
> |Position|82.5|94.58|
>
> These results demonstrate basic generalization from detailed to abstract instructions, yet sill warranting further investigation for smaller models (e.g., low compile rates for 3B models).
>
> Regarding real-world deployment and application, we focus on two primary aspects: compile rate and information extraction metrics. Compile rate reflects the system’s ability to successfully execute code and generate BIM models. Even if minor errors occur during instruction interpretation, these can often be rectified through multi-turn interactions, thus supporting iterative refinement. In contrast, information extraction metrics assess the system’s capacity to accurately identify key parameters and capture the intended structure of MBLs within BIM. High performance on both dimensions suggests that the system is suitable for initial deployment in practical settings. However, it is essential to note that this does not imply the problem is trivial or that the benchmark has been saturated; current pass rates remain low, even when provided with detailed instructions. Moreover, as previously discussed, robustly handling general or abstract user instructions remains an open challenge, underscoring the need for ongoing research to enhance the system’s comprehensiveness and reliability. Therefore, we recognize the importance of extending our framework in future work through: (1) Collaborating with architects and engineers to qualitatively assess generated outputs with diverse detail levels; (2) Conducting systematic studies to evaluate usability and user satisfaction across varying granularities; and (3) Developing curriculum learning strategies and improved input/output modeling to address the challenges of abstract instruction.
>
> In response to this useful comment, we will revise the Experiment section to include new experimental results. We will also clarify the limitations of our current evaluation and explicitly outline future directions in the Discussion section.
>
> **3 Insufficient discussions on related design approaches (W2):**
>
> Previous approaches, such as Text2BIM and Tell2Design, primarily employ coordinate-driven methods, where user instructions are mapped to spatial arrangements or explicit geometric coordinates [1, 2]. While Text2BIM focuses on general building design from conceptual natural language, Tell2Design generates floorplans using explicit spatial coordinates from detailed user inputs. In our study, we intentionally avoid referencing specific coordinate-driven approaches by name in experimental comparisons to emphasize the paradigm-level distinction between coordinate-centric and our high-level, programmatic representation based on our designed code architecture. However, we acknowledge that a more explicit discussion of these works would provide clearer context for our contributions.
>
> AutomaTikZ [3] utilizes TikZ as an intermediate representation for text-to-scientific-graphic generation, which conceptually aligns with our approach of using an intermediate, human-readable code architecture. In our work, we develop our own “TikZ,” a domain-specific programming interface with complete implementation tailored for BIM-based MBL. This interface encapsulates complex operations into abstract classes and functions, facilitating conditional language modeling with LLMs. While both AutomaTikZ and our system leverage code format as an intermediate form, our approach is distinct in its focus on parametric building design and BIM integration, considering multiple design constraints, as opposed to general scientific graphics.
>
> DiagrammerGPT [4] employs a two-stage pipeline comprising planning and generation for diagram synthesis. Our Text2MBL system similarly integrates a planning component, but this is embedded within the code architecture rather than separated as an explicit module. Consequently, the LLM outputs directly correspond to parametric design code with extracted parameters. We appreciate the reviewer’s suggestion and recognize that incorporating a dedicated planning stage could further generalize our system to broader or more abstract input scenarios as discussed in Response 2. We will discuss this as a promising direction for future work.
>
> We sincerely thank the reviewer for these insightful suggestions, which are instrumental in more clearly positioning our work within the rapidly evolving field of text-to-structured design systems. In the final version, we will expand the Introduction and Background sections to explicitly discuss the similarities and distinctions between Text2MBL and other approaches.
>
> [1] Leng et al., Tell2design: A dataset for language-guided floor plan generation, ACL 2023
> [2] Du et al., Text2BIM: Generating building models using a large language model-based multi-agent framework, 2024
> [3] Belouadi et al., AutomaTikZ: Text-Guided Synthesis of Scientific Vector Graphics with TikZ, ICLR 2024
> [4] Zala et al., DiagrammerGPT: Generating Open-Domain, Open-Platform Diagrams via LLM Planning, COLM 2024

---

> > ### Comment · Reviewer_9xp5 · 2025-08-05
> >
> > Dear authors, I have checked the rebuttal and other reviews. I appreciate the authors' detailed response to my questions. I maintain my original positive rating.

---

> > > ### Author Response · Authors · 2025-08-09
> > >
> > > We are thankful for the reviewer's kind words regarding our rebuttal and for the careful consideration of our responses. We truly value the constructive feedback, which have helped us enhance our manuscript.

---

### Official Review · Reviewer_HG8y · 2025-07-03

**Clarity:** 3
**Significance:** 2
**Originality:** 2
**Rating:** 4
**Confidence:** 3

**Summary:**

This paper proposes a framework called Text2MBL, which aims to directly generate executable Building Information Modeling (BIM) code from textual descriptions of modular building layout (MBL) design. The framework leverages an object-oriented code architecture to address the unique challenges posed by the three-tiered hierarchical structure of MBLs (modules, units, and rooms), which traditional 2D layout generation methods struggle to handle. To train and evaluate Text2MBL, the authors created a dataset consisting of paired real-world modular housing project descriptions and BIM layouts, and fine-tuned large language models (from the Qwen2.5 family) to output structured action sequences in code.

**Questions:**

1. Based on the examples provided (e.g., the rectangular modules in Figure 1 and Table 1) and the descriptions of the Module class (e.g., `rectangular geometry`)  and Utils class (e.g., `MidPointForRectangle`, `PointInRectangle`) in Appendix B, the system appears primarily designed for regular geometric shapes such as rectangles. However, real-world architectural designs often include irregular geometries (e.g., curved walls or polygonal modules). Can the current system handle such non-rectilinear geometries? If not, how extensible is the current code architecture to support more complex geometric forms?
2. How can the input burden be lowered to support more natural and conceptual language? The introduction states that the use of text is intended to “facilitate more intuitive and accessible user interaction,” but the current framework (as shown in Figure 1) requires very specific geometric and spatial inputs. Can the model understand more abstract or high-level descriptions, such as “create a spacious living room connected to a kitchen”? If not, how might the system be extended in this direction?
3. The "Text2BMI" in L49 is confusing.

**Ethical Concerns:**

["NO or VERY MINOR ethics concerns only"]

**Final Justification:**

Thanks for the detailed responses and additional experimental results, which have addressed most of my concerns. Therefore, I raised my rating to borderline accept.

**Limitations:**

yes

**Quality:**

3

**Strengths And Weaknesses:**

Strengths:

1. The unique “module-unit-room” hierarchy of MBLs introduces structural constraints and complexity. Text2MBL explicitly targets these issues through its tailored design.
2.  The paper builds a dataset based on real-world modular housing projects, pairing textual descriptions with BIM code. To mitigate data scarcity, both partially and fully synthetic data generation strategies are adopted.

Weaknesses:

1. The paper fails to compare with key related methods, especially **Text2BIM**. It is unclear how Text2MBL extends, improves upon, or addresses challenges not solved by Text2BIM (such as MBL’s inherent hierarchy and constraints).
2. As shown in Figure 1, the current framework requires users to specify highly detailed spatial parameters—such as precise module dimensions (length, width), door positions (e.g., on the south wall, 2950mm from the west side), and alignment configurations (e.g., south wall, 2950mm from the west side). Requires very specific and detailed input, which will not be the preferred form of input for actual users.
3. Although the authors fine-tune multiple variants from the Qwen2.5 family and briefly mention GPT-4.1-mini in the appendix, the paper lacks a more systematic comparison of different large language models on the proposed task.
4. While the introduction explicitly states that “design decision-making is no longer designer-exclusive but user-inclusive” (line 27) and that textual descriptions are intended to “facilitate more intuitive and accessible user interaction” (line 28), the paper includes no user study or usability evaluation (e.g., user satisfaction, interaction efficiency). The Discussion section (Section 5) relegates user-centered evaluation to future work, creating a disconnect between the paper’s stated motivation and its actual methodology.

---

> ### Author Rebuttal · Authors · 2025-07-31
>
> We feel great thanks to the reviewer’s professional and constructive comments. We have carefully summarized the valuable weaknesses and questions identified by the reviewer into the following five areas: (1) level of user interaction detail, (2) method comparison and improvement, (3) motivation and conceptualization, (4) framework extension, and (5) typo rectification.
>
> To enhance understanding, we address each area separately, providing a thorough explanation of the modifications made.
>
> **1 Level of user interaction detail (W2 and Q2):**
>
> Our current framework is grounded in the paradigm of parametric design, wherein the manipulation of explicit spatial and geometric parameters is standard practice during the architectural and structural design phases. This approach is consistent with prior works on floorplan generation from textual descriptions [1], which similarly utilize precise input. Based on this, our work can provide a robust BIM foundation for subsequent stages in the lifecycle of modular buildings—a capability that necessitates detailed and unambiguous specification of components. Therefore, our evaluation emphasizes metrics related to exact code execution and the accuracy of information extraction, ensuring the practical applicability.
>
> We acknowledge, as the reviewer notes, that other approaches focus on high-level room layout or furniture configuration (e.g., through retrieval mechanisms) using more general or abstract user input [2, 3, 4]. However, our current scope remains on parametric design, which we believe is essential for ensuring downstream BIM usability and compliance with industrialized construction standards. We agree that future design tools should integrate both paradigms, and we view our contribution as a foundational step toward this broader goal.
>
> To address the reviewer’s question regarding support for more abstract or conceptual descriptions, we conducted additional experiments to evaluate the framework’s generalization capability. Specifically, we tested the system on vague and high-level instructions (e.g., “Generate a layout with x module, x unit, x living room, x bathroom, x bedroom, x kitchen”). We report the compile rates (i.e., the successful generation of executable code) and extraction F1 scores (measuring the accurate extraction and mention of modules, units, and rooms).
>
> |Model and output|Compile|F1|
> |-|-|-|
> |Qwen2.5-Coder-3B|||
> |Name|68.75|99.05|
> |Position|45|98.36|
> |Qwen2.5-Coder-7B|||
> |Name|92.5|100|
> |Position|95|99.53|
> |Gpt-4.1-mini|||
> |Name|81.25|92.91|
> |Position|78.75|91.96|
> |Gpt-4.1|||
> |Name|87.5|96.11|
> |Position|82.5|94.58|
>
> These results demonstrate initial generalization capabilities of language models trained and further validate the applicability of our framework. We agree with the reviewer that lowering the input burden to support more natural and conceptual language is an important direction for future work. Achieving this will likely require (1) improved input/output modeling at varying granularity, (2) curriculum learning strategies, and (3) systematic user studies to evaluate usability. We are actively exploring these avenues and consider them a priority for future research.
>
> In response to the valuable feedback, we will make the following revisions in our final version. We will first clearly articulate the intention and scope of our work. Then, we will incorporate additional generalization experiments in the Experimental section. The implications of them will be discussed in the Discussion section.
>
> [1] Leng et al., Tell2design: A dataset for language-guided floor plan generation, ACL 2023
> [2] Yang et al., Llplace: The 3d indoor scene layout generation and editing via large language model, 2024
> [3] Fu et al., AnyHome: Open-Vocabulary Generation of Structured and Textured 3D Homes, ECCV 2024
> [4] Aguina-Kang et al., Open-Universe Indoor Scene Generation using LLM Program Synthesis and Uncurated Object Databases, 2024
>
> **2 Method comparison and improvement (W1 and W3):**
>
> We appreciate the reviewer’s emphasis on the need for a thorough comparison. Text2MBL introduces a new paradigm by encapsulating design details within structured code, exposing only function and class interfaces. This enables a code generation approach that inherently captures the hierarchical and constraint-based nature of BIM-based MBL, which is a core challenge in parametric design. In contrast, Text2BIM adopts a coordinate-driven method, directly generating geometric coordinates without modeling deeper design semantics or structural constraints present in MBL, as compared in our manuscript. As discussed in the manuscript, this limitation often necessitates extensive post-processing and human intervention when adapting such outputs to BIM workflows, e.g., careful management of component sequencing and resolving geometric overlaps or inconsistencies. By comparison, Text2MBL produces outputs that are immediately compatible with BIM environments and can be seamlessly integrated into real-world design workflows, significantly reducing the need for additional processing and facilitating practical deployment during the design phase. In our work, we omit naming specific coordinate-driven methods (e.g., Text2BIM or Tell2Design) in experimental comparisons to highlight the fundamental paradigm difference between coordinate- and code-based approaches.
>
> We acknowledge the reviewer’s suggestion for a more comprehensive evaluation of different LLMs. In response, we have expanded our experiments to include more models. The comparative results are summarized in the table below:
>
> |Model and output|Compile|Pass|Instance F1|Argument F1|IoU|
> |-|-|-|-|-|-|
> |Gpt-4.1-mini||||||
> |Name|90|6.25|68.89|83.62|88.4|
> |Potision|93.75|3.75|68.33|84.37|94.8|
> |Coordinate|-|-|-|-|27.06|
> |Gpt-4.1||||||
> |Name|97.5|22.5|78.54|90.49|90.03|
> |Potision|93.75|23.75|76.67|88.52|88.95|
> |Coordinate|-|-|-|-|29.94|
> |Gemini-2.5-flash||||||
> |Name|95|21.25|73.58|83.49|84.81|
> |Potision|92.5|7.5|65.13|75.61|88.47|
> |Coordinate|-|-|-|-|23.94|
> |Gemini-2.5-pro||||||
> |Name|97.5|36.25|84.18|88.57|90.24|
> |Potision|98.75|35|84.95|92.02|93.86|
> |Coordinate|-|-|-|-|30.72|
>
> In line with our preliminary findings, we observe that while the model seldom generates fully correct code, it exhibits a notable capacity to extract accurate information from unstructured textual inputs. However, generating correct coordinate sequences remains particularly challenging. This suggests that precise spatial reasoning and the reliable translation of geometric details continue to pose significant limitations for current proprietary models in this domain.
>
> To address the points outlined above, we will thoroughly revise the Experiment section to provide a clearer distinction between the different paradigms employed. Additionally, we will expand Appendix F to present a more comprehensive set of results for various LLMs.
>
> **3 Motivation and conceptualization (W4):**
>
> Our study is situated within the paradigm of parametric design, where users provide detailed textual inputs specifying precise design requirements. Accordingly, our evaluation methodology primarily focuses on the accurate comprehension and execution of these instructions, measured through objective, quantitative metrics such as code correctness and information extraction accuracy.
>
> We fully acknowledge that as the system evolves to handle more conceptual or ambiguous user inputs—such as general, natural language utterances—a comprehensive user study will become essential. Such studies would assess user satisfaction, interaction efficiency, and overall usability, providing a richer understanding of user experience in more open-ended scenarios. We agree that this is a critical direction, and as noted in Response 1, we have provided some extended experiments and identified it as important future work.
>
> In response, we will revise and extend both the Introduction and Discussion sections to better clarify our motivation and elaborate on future research directions informed by our findings.
>
> **4 Framework extension (Q1):**
>
> Our work focuses on modular building layouts, where the modules are rectangular. This design choice aligns with emerging trends in the AEC sector, i.e., industrialized construction. Rectangular modules are favored due to principles such as design for manufacturing and assembly, as well as efficient transportation, which are key considerations in modular construction. As a result, our study and corresponding codebase are tailored to buildings composed of rectangular modules, which currently represent the majority of real-world modular construction projects.
>
> We appreciate the reviewer’s interest in the generalizability of our system. The code architecture is developed with the principle of low coupling and high cohesion, ensuring that each class has a clear, distinct responsibility. Each class and function were completely implemented with full details to allow deployment and application. This separation of concerns facilitates extensibility. For instance, while the current Module and Utils classes focus on rectangular modules, our architecture allows for the integration of additional geometric representations. For non-rectilinear geometries (e.g., curved or polygonal modules), one could extend the Module class by introducing new subclasses (e.g., PolygonalModule or CurvedModule) and corresponding utility methods (e.g., PointInPolygon). These new classes can define their geometric descriptors and methods without interfering with the existing rectangular implementation.
>
> To reflect the points raised above, inspired by the reviewer’s constructive suggestions, we will expand the method description section to provide a more detailed explanation of the potential extensions of our proposed framework.
>
> **5 Typo rectification (Q3):**
>
> We have carefully re-examined the manuscript to ensure clarity and consistency throughout. For example, “Text2MBL” should be inserted as mentioned by the reviewer.

---

### Note · Authors · 2025-08-12

**Dear AC and Reviewers,**

The authors sincerely thank the AC and reviewers for their constructive and insightful feedback. In this final response, we summarize the strengths noted in reviews, address concerns raised, and outline planned revisions.

**Strengths**

**1. Domain-grounded problem formulation** – A systematic study of real-world modular building layouts (MBLs), capturing hierarchical module–unit–room relationships grounded in practice.

**2. New Text-to-BIM task** – Generating MBLs directly in Building Information Modeling (BIM) environments from natural language via code generation, bridging language understanding and design modeling.

**3. Dataset curation** – Real modular housing project data paired with BIM code and descriptions, supplemented by partially/fully synthetic data to address scarcity.

**4. End-to-end system implementation** – A fully implemented BIM-based system using secondary development tools for automatic  MBL generation in BIM.

**5. Comprehensive experimental evaluation** – Extensive experiments and error analyses revealing both strengths and limitations.

**Concerns and Responses**

**1. Lack of background information** - We have expanded BIM concepts, standards, and practical relevance, and added comparisons to existing design tools.

**2. Motivation and conception** - We have clarified the target scope (parametric design in BIM), refined the problem statement, and emphasized practical impact.

**3. Framework presentation** - We have added clearer, structured explanations and highlighted extensibility to other design scenarios.

**4. Generalization capability** - We have added experiments with abstract instructions, showing moderate robustness and outlining future improvements.

**5. Zero-shot comparison** - We have included evaluations with multiple LLMs on both code-based and coordinate-based representations; results favor our code-based formulation, underscoring dataset utility.

**Planned Revisions**

1. Revise *Introduction/Background* for motivation, scope, BIM, and related works.

2. Enhance *Text2MBL* with more framework explanations and new illustrations.

3. Expand *Experiment* with generalization and zero-shot results (Appendix F).

4. Update *Discussion* with new insights on generalization and applicability.

We thank the AC and reviewers again for their valuable and thoughtful feedback. We believe these revisions will significantly improve the clarity, technical depth, and accessibility of our work.

---

### Decision · Program_Chairs · 2025-09-17

**Decision:**

Accept (poster)

**Comment:**

This paper introduces Text2MBL, a framework for generating Building Information Modeling (BIM) code from textual descriptions of modular building layouts (MBLs). The core idea involves using an object-oriented code architecture tailored to the hierarchical structure of MBLs (modules, units, rooms) and fine-tuning large language models (LLMs) to output structured action sequences in code. A dataset of paired descriptions and ground truth layouts was curated from real-world projects to train and evaluate the framework.

Strengths: The reviewers consistently highlight the novelty of the task formulation, which bridges natural language understanding and structured architectural modeling. The comprehensive system implementation in Autodesk Revit and the curation of a paired dataset of real and synthetic MBLs are also praised. The code-driven generation method consistently outperforms coordinate-based baselines, as demonstrated by the strong empirical results.

Weaknesses: A major concern is the limited comparison with related LLM-based design tools, especially Text2BIM and other relevant works in text-to-scene generation. Some reviewers point out that the current framework requires highly detailed spatial parameters as input, which may not be preferred by actual users. The lack of external validation through user studies or domain expert evaluation is also noted. The relative ease of the benchmark, suggested by the high scores, is questioned, with some reviewers wondering if the task is too easy and whether the model is close to real-world deployability.

The authors addressed many of the reviewers' concerns during the rebuttal phase. They provided additional experimental results, including zero-shot evaluations and experiments with more abstract instructions. They clarified the distinctions between their work and related approaches, and they also detailed the potential extensions of their framework. The reviewers generally acknowledged that the authors adequately addressed their concerns, and some reviewers raised their scores accordingly.

Overall, this paper presents a technically sound and well-executed approach to a novel and relevant problem. While the limitations regarding comparisons to other methods and the need for more user-centric evaluation are valid, the strengths of the task formulation, system implementation, and empirical results outweigh these concerns. Therefore, the AC recommends accepting this paper.